# Liquid-like versus stress-driven dynamics in a metallic glass former observed by temperature scanning X-ray photon correlation spectroscopy

Maximilian Frey [1] ✉, Nico Neuber [1], Sascha Sebastian Riegler [1], Antoine Cornet[2,3], Yuriy Chushkin[3], Federico Zontone[3], Lucas Matthias Ruschel [1], Bastian Adam [1], Mehran Nabahat[4], Fan Yang [5], Jie Shen[2,3], Fabian Westermeier [6], Michael Sprung [6], Daniele Cangialosi [7,8], Valerio Di Lisio [7], Isabella Gallino[9], Ralf Busch [1], Beatrice Ruta [2,3] & Eloi Pineda [4]

Since several decades, the dynamics and vitrification kinetics of supercooled liquids are the subject of active research in science and engineering. Profiting from modern detector technology and highly brilliant fourth-generation synchrotron radiation, we apply temperature scanning X-ray photon correlation spectroscopy (XPCS) to probe the dynamics of a Pt-based metallic glass former in the glass, glass transition region, and supercooled liquid, covering up to six orders of magnitude in timescales. Our data demonstrates that the structural α-relaxation process is still observable in the glass, although it is partially masked by a faster source of decorrelation observed at atomic scale. We present an approach that interprets these findings as the superposition of heterogeneous liquid-like and stress-driven ballistic-like atomic motions. This work not only extends the dynamical range probed by standard isothermal XPCS but also adds a different view on the α-relaxation across the glass transition and provides insights into the anomalous, compressed temporal decay of the density-density correlation functions observed in metallic glasses and many out-of-equilibrium soft materials.

When a liquid is undercooled, it changes from a stable to a metastable equilibrium state. Its structural dynamics undergo a drastic slowdown over several orders of magnitude. The temperature dependence of this slowdown can differ significantly among different systems, as described by Austen Angell's famous fragility concept[1] that categorizes liquid dynamics by their deviation from an exponential, Arrhenius-like temperature dependence. The dynamic slowdown of undercooled liquids is the precondition that eventually results in vitrification, where the liquid

[1]Chair of Metallic Materials, Saarland University, Saarbrücken, Germany. [2]Institut Néel, Université Grenoble Alpes and Centre National de la Recherche Scientifique, Grenoble, France. [3]European Synchrotron Radiation Facility, Grenoble, France. [4]Department of Physics, Institute of Energy Technologies, Universitat Politècnica de Catalunya—BarcelonaTech, Barcelona, Spain. [5]Institut für Materialphysik im Weltraum, Deutsches Zentrum für Luft- und Raumfahrt (DLR), Köln, Germany. [6]Deutsches Elektronen-Synchrotron DESY, Hamburg, Germany. [7]Donostia International Physics Center, San Sebastián, Spain. [8]Centro de Física de Materiales (CSIC-UPV/EHU), San Sebastián, Spain. [9]Metallic Materials, Technical University of Berlin, Berlin, Germany. ✉e-mail: maximilian.frey@uni-saarland.de

'falls out of equilibrium'[1] and forms a non-ergodic, non-equilibrium glass. In other words, liquid dynamics predetermine vitrification kinetics. Hence, understanding the nature of the dynamic slowdown and fragility is of fundamental relevance for glass science and industry-relevant aspects like e.g., the glass forming ability of a given system.

A mighty tool to determine the dynamics of disordered materials is X-ray photon correlation spectroscopy (XPCS), a time-resolved experimental approach[2] that uses monochromatic and coherent synchrotron radiation. Coherent diffraction on amorphous matter creates a speckle pattern that reflects the configuration of the microscopic ensemble in the probed sample volume. Temporal changes in the microscopic configuration induce changes in the speckle pattern and accordingly, the correlation between two speckle patterns will decay with growing temporal distance. Analyzing the intensity autocorrelation function thus allows us to determine microscopic relaxation times. XPCS studies that resolve the atomic-scale dynamics of metallic glass formers require highly brilliant radiation to compensate the weak scattering signal of these amorphous materials and are therefore only possible since about one decade[3]. While the signal-to-noise ratio and detector technology limited the exposure time to several seconds in earlier studies[3–10], recent advances allowed exposure times in the subsecond regime[11–15].

While some non-isothermal XPCS approaches can be found in literature[15–18], XPCS studies on metallic glass formers are usually measured under isothermal conditions. In the present study, we instead perform temperature scanning XPCS, where the temperature is changed continuously with a constant rate. In order to do so, we use the high flux available at the ID10 beamline of the fourth-generation European Synchrotron Radiation Facility (ESRF) after the Extremely Brilliant Source (EBS) upgrade[19,20]. The test subject of this work is the $Pt_{42.5}Cu_{27}Ni_{9.5}P_{21}$ alloy[21]. Its excellent thermal stability against crystallization previously allowed our groups to investigate the supercooled liquid (SCL) dynamics through isothermal XPCS experiments[13]. The present work extends these studies by slowly heating and cooling the sample through the glass, the glass transition, and the SCL state.

In the equilibrium SCL, the decay of the obtained intensity autocorrelation functions can be modeled accurately by a conventional Kohlrausch-Williams-Watts (KWW) equation with stretched shape, which reflects the heterogeneous, liquid-like atomic motions of the α-relaxation process, as previously observed by isothermal XPCS[13]. Furthermore, the temperature dependence of the determined relaxation times is found to mimic the alloy's fragility measured by macroscopic viscosity measurements[22]. A different picture arises in the non-equilibrium state, i.e., in the glass and the glass transition region. Here, the conventional KWW fit fails to describe the decay of the intensity autocorrelation functions. Instead, we successfully model the complexly shaped decorrelation through a multiplication of two KWW functions. One of these features a stretched shape (KWW shape exponent <1), while the other one features a compressed shape (KWW shape exponent >1).

Based on the long-standing concept of spatially heterogeneous dynamics, we offer a scenario that explains these findings through two different but simultaneously occurring types of atomic motion. More precisely, we interpret the vitrification process as the interlocking of nm-scale regions with slower dynamics, so termed 'rigid domains'. This jamming-like effect results in volumetric frustration and internal stress gradients, which induce ballistic-like 'microscopic drift' movements with a typical compressed decorrelation footprint. On the other hand, we attribute the stretched decorrelation component to atomic motions related to the liquid-like α-relaxation, which remains partially active in the non-equilibrium. The superposition of these two types of atomic motion finally creates the complex decorrelation shape observed in the non-equilibrium states.

## Results

### A first overview of the temperature scanning XPCS results

The thermal program applied during the XPCS experiments consists of a heating scan and a subsequent cooling scan, both performed with a rate of $0.0167\,K\,s^{-1}$ (1 K min$^{-1}$). It is retraced ex-situ by differential scanning calorimetry (DSC) and the obtained heat flow signal is shown in Fig. 1A. The initial heating of the as-spun ribbon sample starts in the glassy state, passes the glass transition region between 500 K and 525 K, and enters the SCL state until the maximum temperature of 548 K is reached. In the cooling scan, the sample is cooled until vitrification occurs between 521 K and 490 K, finally reentering the glassy state. Accordingly, the thermal ranges of the glass, glass transition, and SCL regions are distinguished by gray, light gray, and white backgrounds to guide the eyes in Fig. 1A.

Two time correlation functions (TTCFs) and intensity autocorrelation functions, $g_2$, are evaluated in subsequent batches of 240 s duration, as further explained in the "Materials and Methods" section. To provide an overview, we focus on six of these evaluation batches in the following, numbered from B1 to B6. They are located at representative temperatures in the glass (480 K), glass transition (510 K), and SCL (541 K) upon heating and cooling, as indicated in Fig. 1A. The corresponding TTCFs are shown in Fig. 1B to give a first graphical impression of the evolution of dynamics during the temperature scanning procedure. Warm colors depict high intensity-intensity correlation between two experimental times $t_1$ and $t_2$, whereas cold colors describe low correlation. As expected, decorrelation accelerates with rising temperature, which is graphically represented by a clear narrowing of the warm-colored stripe along the diagonal of the TTCFs. The $g_2$ data in Fig. 1C, D mirrors this trend by showing faster decays at higher temperatures, indicating an overall acceleration of dynamics due to increased thermal motion.

### Conventional KWW fitting in the equilibrium

In liquids, the temporal decay of the intermediate scattering function (ISF) and the related $g_1$ function is usually described by a KWW function[23]

$$|g_1(t)| = f\exp\left(-\left(\frac{t}{\tau}\right)^{\beta}\right) \quad (1)$$

and since $g_1$ and $g_2$ are connected through the Siegert relation[24,25],

$$g_2(t) = b + c\,|g_1(t)|^2 \quad (2)$$

it follows that the intensity autocorrelation function can be modeled as

$$g_2(t) = b + c\exp\left(-2\left(\frac{t}{\tau}\right)^{\beta}\right) \quad (3)$$

where f is the non-ergodicity parameter, τ is the relaxation time, β is the shape exponent, c is the experimental contrast, and b is the baseline value. The resulting KWW fits of the $g_2$ data are displayed by black solid curves in Fig. 1C, D. In the SCL state (B3 and B4), the KWW function models the stretched decay of $g_2$ quite satisfactorily, in agreement with previous findings[6,12,13,26]. A different picture arises in the non-equilibrium, i.e., in the glass transition and glass region (B1, B2, B5, and B6), where the KWW fits clearly fail to describe the complex shape of $g_2$. More precisely, the initial parts of the $g_2$ data, roughly the first three decades in timescale, from 0.01 to 10 s, behave more stretched than the overall KWW fit curve, while the later parts appear more compressed. Hence, we observe a kind of 'cut-off' behavior, where an initially stretched decay abruptly changes into a more compressed appearance.

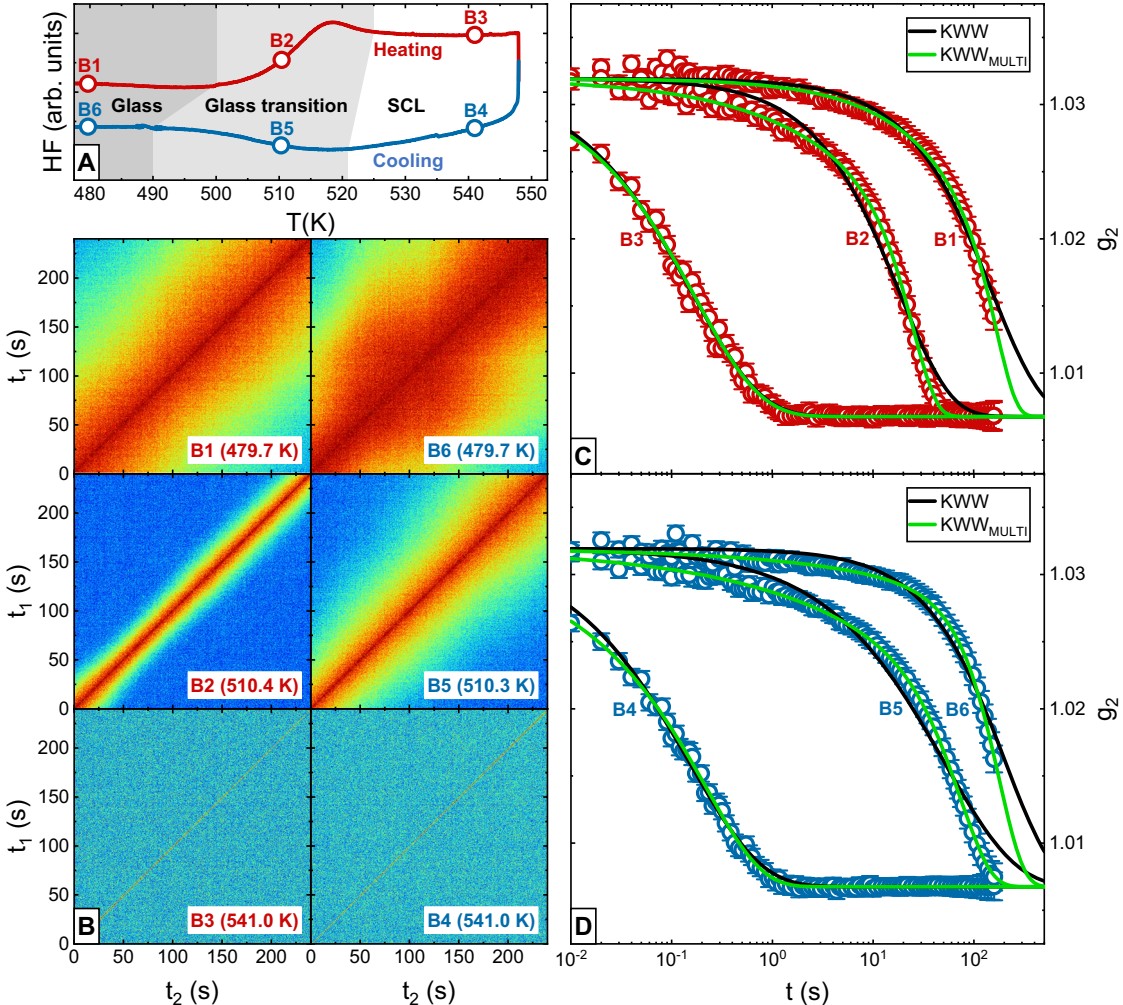

**Fig. 1 | Applied thermal program, representative TTCFs and $g_2$ curves, as well as different fitting approaches. A** Ex-situ differential scanning calorimetry (DSC) scans retrace the 0.0167 K s$^{-1}$ heating-cooling protocol applied for the X-ray photon correlation spectroscopy (XPCS) measurements. The three thermal regions, namely glass, glass transition, and supercooled liquid (SCL), are distinguished through shaded backgrounds. Furthermore, circles highlight the temperatures of six representative $g_2$ evaluation batches, numbered from B1 to B6. **B** The two time correlation functions (TTCFs) of these six $g_2$ batches illustrate the accelerating dynamics with increasing temperature. Each TTCF includes information stemming from 24000 frames. For their respective color bars, see Supplementary Fig. 9 in the SI. The corresponding intensity autocorrelation functions are further given in (**C, D**). Each $g_2$ data set consists of 96 mean value data points, the error bars represent standard deviations. The Kohlrausch-Williams-Watts (KWW) fits and KWW$_{MULTI}$ fits are shown as black and green lines, respectively. Both fit approaches provide satisfying and similar results in the SCL state (B3 and B4), but while the KWW approach fails to describe the shape of the $g_2$ decorrelation in the glass and especially in the glass transition region, KWW$_{MULTI}$ provides meaningful results (B1, B2, B5, and B6). The fit parameters of the shown KWW and KWW$_{MULTI}$ curves can be found in Fig. 3.

## On the multiplicative KWW fitting used in the non-equilibrium

Previous XPCS studies revealed that metallic glass formers show an anomalous crossover in their microscopic dynamics. While the metastable equilibrium SCL phase typically features stretched decorrelation with $\beta$ values below 1[6,12,13,26,27], compressed decorrelation with $\beta$ values above 1 is found in the non-equilibrium glass[3,5,9,12,14,26,28–31]. Yet, a failure of the KWW model due to the complex superposition of stretched and compressed components as observed in the present work is, to our best knowledge, a feature that has not been reported so far. We attribute this observation to the fast exposure time of 0.01 s, which allows us to explore almost five orders of magnitude in timescale and to observe the glass transition upon temperature scanning as a gradual crossover between liquid-typical stretched and glassy-typical compressed decay. Earlier XPCS studies at third generation synchrotrons were technically limited to exposure times in the order of seconds[3,5,7,9,10,26,28,30,32–34], therefore lacking the temporal resolution to detect such features. It shall be furthermore mentioned that macroscopic approaches like e.g., stress relaxation measurements usually do not feature compressed decays[35–37], albeit exceptions were recently reported[38].

The complex decorrelation behavior in the non-equilibrium asks for an adapted fitting approach. In general, the temporal decorrelation of the ISF results from all the relative positional changes $\Delta \mathbf{r}_{j-k}(t) = \mathbf{r}_j(t) - \mathbf{r}_k(0)$ between all scattering particles N that occur within the time coordinate t[39–42]:

$$g_1(t) \propto \left\langle \sum_{j=1}^{N} \sum_{k=1}^{N} a_j(0)\, a_k(t) \exp\left(-i\boldsymbol{q}\, \Delta \mathbf{r}_{j-k}(t)\right) \right\rangle \quad (4)$$

where $a_j$ and $a_k$ are the scattering factors of the respective particles. SCLs are well-known to feature heterogeneous dynamics, which implies large spatial and temporal fluctuations in $\Delta \mathbf{r}_{j-k}(t)$[43–45]. Hence, the ISF decorrelation incorporates a broad distribution $D(\tau')$ of exponential relaxations $g_{1,\tau'}(t) = \exp(-t/\tau')$ extending over several

decades according to

$$g_1(t) = \int g_{1,\tau'}(t)\, D(\tau')\, d\tau' \tag{5}$$

and eventually, this leads to the liquid-typical stretched exponential shape[43–45].

Yet, compressed decorrelation cannot be explained by a distribution of exponential contributions. In this case, some models have demonstrated that stress-driven particle rearrangements can produce a compressed decay of the ISF[46]. Our ansatz will be the following: In the non-equilibrium state, the scattering particles can be subjected to two different types of motion, one liquid-like leading to a stretched decorrelation component (index S) and another one resulting in a compressed decorrelation component (index C), which is activated once the system is rigid enough to build the necessary stresses.

The relative positional changes of particles, therefore, result from a sum of these two types of particle movement, $\Delta\mathbf{r}_{j-k}(t) = \Delta\mathbf{r}_{j-k,S}(t) + \Delta\mathbf{r}_{j-k,C}(t)$. Hence, it follows from Eq. 4 that the global $g_1$ function can be approached by a factorization[41,47–49] as

$$\left|g_1(t)\right|^2 = \left|g_{1,S}(t)\right|^2 \left|g_{1,C}(t)\right|^2. \tag{6}$$

Regarding the Siegert relation in Eq. 2, a multiplicative KWW fit function unfolds for the observed $g_2$ decay, termed KWW$_{\text{MULTI}}$ in the following:

$$g_2(t)_{t_{batch}} = b + c \exp\left(-2\left(\frac{t}{\tau_S}\right)^{\beta_S}\right) \exp\left(-2\left(\frac{t}{\tau_C}\right)^{\beta_C}\right) \tag{7}$$

The KWW$_{\text{MULTI}}$ fits of the six representative evaluation batches are shown in Fig. 1C, D as green solid lines. In the SCL state (B3 and B4), no significant increase in fit quality is gained by changing from conventional KWW to KWW$_{\text{MULTI}}$, since both approaches create practically overlapping curves. This changes drastically in the glass transition region (B2 and B5), where KWW$_{\text{MULTI}}$ provides adequate fitting while the conventional KWW fails to do so. In the glassy state (B1 and B6), the misfit between $g_2$ data and KWW fit might be less distinct than in the glass transition region, but still, KWW$_{\text{MULTI}}$ allows for more accurate fitting of the data.

Figure 2 focuses on batch B5 located in the glass transition region upon cooling to provide an in-depth comparison of the suboptimal KWW fit (black curve) and the well-functioning KWW$_{\text{MULTI}}$ (green curve) approach. The orange and purple dashed curves depict the two components of the KWW$_{\text{MULTI}}$ fit, KWW$_S$ as well as KWW$_C$. Here, the rather fast ($\tau = 133\,\text{s}$), but also highly compressed ($\beta = 1.61$) KWW$_C$ provides almost no decorrelation in the first three decades below 10 s. This leads to the apparently counterintuitive result that the short-time domain is rather dominated by the slow ($\tau = 3423\,\text{s}$), but quite stretched ($\beta = 0.33$) KWW$_S$ (orange arrow and background). It is only after the initial 10 s that the compressed KWW$_C$ gains momentum. After 57 s, it overcomes KWW$_S$ and starts to dominate the overall $g_2$ decay, allowing to describe the cut-off appearance (purple arrow and background). Here, it shall be mentioned that the factorization in Eq. 6 is strictly valid only if the timescales of the two motion types are sufficiently different from each other[47] and, therefore, one of the two processes clearly dominates the decay. Hence the KWW$_{\text{MULTI}}$ model will be an accurate approach in the short time region, the first three decades in Fig. 2, where only the fast time portion of the broad relaxation time distribution underlying the stretched decay contributes to the heterogeneous ISF. In the rather narrow timescale region dominated by the compressed cut-off decorrelation, the two sources of motion cannot be precisely disentangled, and the multiplicative approach should be interpreted as a functional shape that can

adapt to model the structural dynamics participated by both sources of motion.

For the sake of completeness, we want to note that a compressed decay in $g_2$ can also stem, in principle, from a macroscopic sample movement relative to the incoming photon beam, called transit decorrelation[41,50]. Indeed, the constant temperature change of the temperature scanning method implies a corresponding thermal expansion of the measurement setup, which could cause such an artifact. Yet, we estimated this effect based on the conditions of the given experimental setup and found it to be negligible, as further explained in the Supplementary Information (SI).

**Comparing conventional and multiplicative KWW fitting results**
Introducing additional fit parameters, and, therefore, additional degrees of freedom, to a fit function easily improves its adaptivity towards a given data set. To validate the physical meaning behind the fit results, Fig. 3 compares $\tau$ and $\beta$ values obtained from conventional KWW and KWW$_{\text{MULTI}}$ during both, heating and cooling. Again, the ex-situ DSC scans are provided, see Fig. 3A, D, and the thermal ranges of the glass, glass transition, and SCL regions are separated by gray, light gray, and white backgrounds. For reasons of clarity, fit results are only displayed in the temperature regions where they appear meaningful. Hence, the KWW results are only shown in the SCL state, while the KWW$_{\text{MULTI}}$ data is limited to the glass and glass transition region. Nevertheless, the complete data sets over the full temperature range can be found in Supplementary Fig. 2 in the SI.

Upon heating in the glass and through the full glass transition region, $\tau_C$ in Fig. 3B features a glass-typical weak temperature dependence in agreement with previous works[3,42]. While $\tau_S$ is roughly in the order of $\tau_C$ at lower temperatures in the glass, it departs from $\tau_C$ at about 480 K and starts to rise. We interpret this as aging behavior, the onset of major structural rearrangements in the proximity of the glass transition and the consequent relaxation of the system towards the SCL upon heating with a rate much slower than the fast quenching used to produce the as-spun glass[3,42]. The DSC scan in Supplementary Fig. 3 in the SI further supports this interpretation. It shows a massive exothermal signal below the glass transition, which is a well-known indication for aging. $\tau_S$ reaches a maximum in the initial part of the calorimetric glass transition at about 506 K and then adopts a steep negative temperature dependence, hence, approaching a liquid-like trendline. When entering the SCL, $\tau_C$ and $\tau_S$ merge into the conventional KWW $\tau$ curve, which continues the liquid-like temperature trend previously seen for $\tau_S$ to finally reach values as fast as 0.2 s at the maximum temperature. The shape exponents $\beta_C$ and $\beta_S$ depart drastically in the glass and glass transition region. With values in the order of 2, $\beta_C$ is highly compressed, but abruptly drops to values around unity when reaching the SCL state. In contrast, $\beta_S$ features stretched values between 0.35 and 0.7 in the whole observed temperature range. In the glass, it shows a slight downwards trend with rising temperature. Yet, at 506 K, the temperature at which $\tau_S$ enters the liquid-like trendline, $\beta_S$ also starts to feature a faint upwards trend, which is in agreement with previous data obtained in the SCL by Neuber et al.[13] This temperature behavior of $\beta_S$ also resembles the one observed in macroscopic stress relaxation dynamics[35]. As an additional verification, KWW and KWW$_{\text{MULTI}}$ fitting were also tested with fixed $\beta$ and $\beta_S$ values to decrease the number of parameters in the fitting routine and improve the level of confidence. Although the general interpretation is not affected, these approaches provided less optimal results as they do not take into account the small but relevant temperature trends of these shape exponents in the probed $Pt_{42.5}Cu_{27}Ni_{9.5}P_{21}$ alloy[13] (see also Supplementary Fig. 8 and the related discussion in the SI).

Regarding the results of the cooling scan in Fig. 4E, F, the $\tau$ and $\beta$ values obtained in the SCL state basically show the similar temperature trends as during heating. When entering the glass transition region, $\tau_C$ and $\tau_S$ again decouple drastically. While $\tau_C$ changes to a glass-typical,

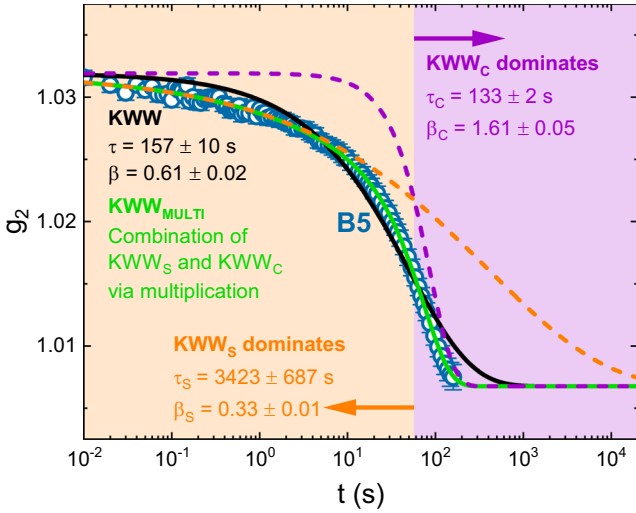

**Fig. 2 | The KWW$_{MULTI}$ fitting approach in detail.** Intensity autocorrelation function of the representative batch B5, located in the glass transition region upon cooling (510.3 K) as indicated in Fig. 1. The data set consists of 96 mean value data points, the error bars give the standard deviations. The black solid curve represents the conventional Kohlrausch-Williams-Watts (KWW) fit, which fails to describe the cut-off shape of the $g_2$ decorrelation. Instead, the KWW$_{MULTI}$ fit (green solid line) describes the decay exceptionally well. It results from a multiplication of a stretched (KWW$_S$) and a compressed (KWW$_C$) component (orange and purple dashed lines). While KWW$_S$ dominates the KWW$_{MULTI}$ fit roughly within the first 57 s (orange background and arrow), KWW$_C$ dominates at longer timescales (purple background and arrow), thereby creating the apparent cut-off appearance.

nearly temperature-insensitive course, $\tau_S$ follows the liquid-like trendline until the end of the glass transition at about 490 K, where it finally leaves the equilibrium course at high values in the order of $10^5$ s. The decoupling is simultaneously observed in the shape exponents. $\beta_C$ rises to highly compressed values of about 2, while $\beta_S$ keeps slightly decreasing with temperature until reaching values as low as 0.3 when the glass is approached. Comparably low values of the stretching exponent in well relaxed metallic glasses are found in macroscopic relaxation dynamics and simulations[36,37].

Overall, the astonishingly good temperature agreement between calorimetric signals and changes in the KWW and KWW$_{MULTI}$ fitting parameters shall be emphasized, especially concerning the thresholds between glassy state, glass transition region, and SCL. These findings speak in favor of the applied temperature correction procedure, which is described in the "Methods" section and in the SI.

### Additive KWW fitting as an alternative in the non-equilibrium

It is important to stress that the presence of multiple dynamical contributions in the ISF can be modeled also with a sum of two KWW expressions (further termed KWW$_{ADD}$) with comparable accuracy, and that the discrimination between KWW$_{ADD}$ and KWW$_{MULTI}$ is not straightforward. In literature, KWW$_{ADD}$ is usually employed when the timescales associated with the two processes are sufficiently separated, resulting in a split of the correlation curves into two distinctly separated parts and creating a step-like appearance in the decay[15,51-53] Yet, this is not observed at the atomic length scales probed in our present experiments, since we only see (cut-off) single decays as shown in Fig. 1 and Fig. 2. By its mathematical construction, KWW$_{ADD}$ therefore locates the timescale of the stretched component ($\tau_{S,ADD}$) at nearly equal or shorter times than found for the compressed component ($\tau_{C,ADD}$), see Supplementary Fig. 7 in the SI. Consequently, the main difference between KWW$_{ADD}$ and KWW$_{MULTI}$ is based on the temperature evolution of $\tau_{S,ADD}$, which features rather unphysical

activation energies in the non-equilibrium glass transition and glass region. As discussed in detail in the SI, we therefore believe that the multiplicative model is more appropriate to describe the current data.

### A comparison with macroscopic fragility measurements

An option to validate the temperature scanning XPCS method is to evaluate if it arrives at the same SCL fragility as determined by other experimental approaches. For this purpose, Fig. 4 compares the present KWW-fitted $\tau$ results to their equivalents from isothermal XPCS by Neuber et al.[13] as well as to equilibrium viscosities stemming from thermomechanical analysis (TMA) by Gross et al.[22]. To allow a direct comparison in the same physical quantity, the relaxation times are converted into viscosity values, $\eta(\tau)$, using the Maxwell relation $\eta = G\,\tau$[54] ($G$ being a shear modulus) and the cooperative shear model (CSM)[55,56], as explained in detail in the "Methods" section. The data sets show broad agreement and obey the same CSM fit curve (black line), demonstrating their basically identical temperature dependence, i.e., fragility. This demonstrates that temperature scanning XPCS can provide substantial results on par with isothermal XPCS[13] and macroscopic approaches like the here shown TMA[22] or also calorimetry[22,57].

The $\tau_S$ data from the KWW$_{MULTI}$ fitting are also converted into viscosities, $\eta(\tau_S)$, and aligned in Fig. 4. $\eta(\tau_S)$ follows the CSM fit and the course of the TMA equilibrium viscosities throughout most of the glass transition region upon heating and cooling, as indicated by the arrows. This behavior confirms a trend already observed in Fig. 3, which is the KWW$_S$ component extrapolating the liquid-like dynamics from the equilibrium SCL state into the non-equilibrium glass transition region. During cooling, signs of vitrification can only be observed in the glass region, where $\tau_S$ leaves the equilibrium trendline. Upon heating, the KWW$_S$ component reflects aging of the relatively unstable as-spun glass, as $\tau_S$ gradually evolves towards the equilibrium trendline. Hence, KWW$_S$ can be seen as the KWW$_{MULTI}$ component that thaws first during heating and freezes last during cooling.

Finally, it shall be noted that instead of using the KWW relaxation times, the comparison between equilibrium viscosity and XPCS results can be also validly established using the average relaxation time $<\tau>$[58], see Eq. (9) in the "Methods" section. This property combines the $\tau$ and $\beta$ parameters from the KWW functions into a timescale parameter that also includes information about the shape of the decorrelation and hence, about the relaxation time distribution $D(\tau')$. We see in Fig. 4 that $\eta(<\tau>)$ and $\eta(<\tau>_S)$ behave practically identical to $\eta(\tau)$ and $\eta(\tau_S)$, as the respective data sets show large overlap.

### Discussion

The shape exponents $\beta$ and $\beta_S$ in Fig. 3C, F feature values distinctly below unity that reflect the stretched exponential decay typical for the heterogeneous nature of supercooled metallic glass forming liquids[6,12,13,26]. Furthermore, $\beta$ and $\beta_S$ decrease with decreasing temperature, as previously observed in the isothermal XPCS studies by Neuber et al.[13]. While the origin of this well-known temperature trend in the shape exponent is still subject of vital debates[59], it is often correlated with liquid dynamics becoming more temporally[60-62] and spatially[44,45,62] heterogeneous with undercooling. While the former describes a general, non-localized tendency towards a broadening relaxation time distribution, $D(\tau')$ (see Eq. 5), the latter specifically refers to spatial fluctuations in the dynamics[45,63]. Voyles et al. recently used novel electron correlation spectroscopy (ECS)[64] to image such spatio-temporally heterogeneous dynamics in supercooled $Pt_{57.5}Cu_{14.7}Ni_{5.3}P_{22.5}$, an alloy similar to our present system[65,66]. Large differences in relaxation time of up to two orders of magnitude were observed between neighboring nm-sized domains, thus, on a length scale that is typical for the medium range order (MRO). This implies a sub-diffusive[60,67,68] structural relaxation process that is governed by cooperative atomic rearrangements and caging effects.

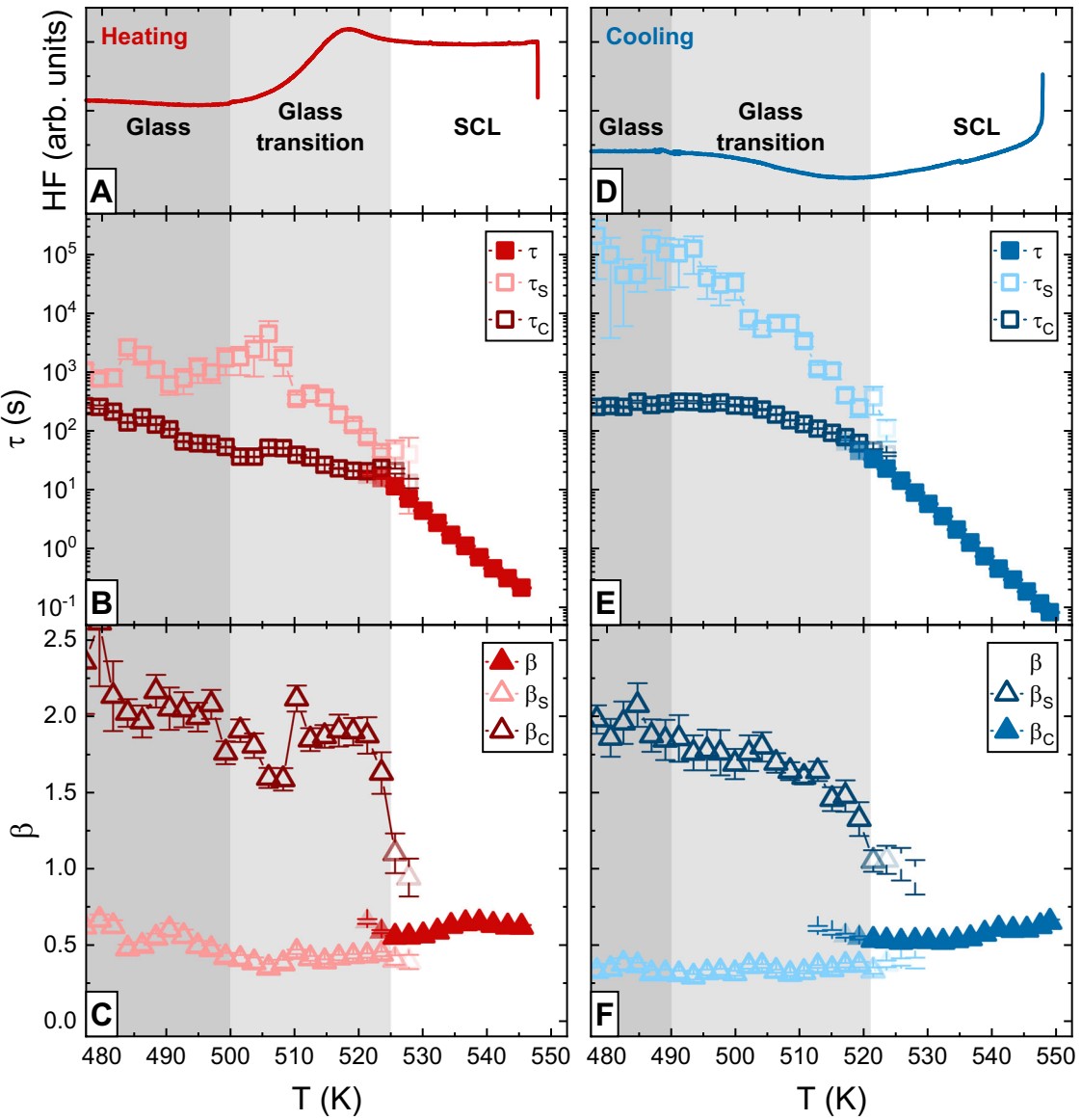

**Fig. 3 | KWW and KWW$_{MULTI}$ fit results.** Reference differential scanning calorimetry (DSC) scans for heating and cooling are provided in (**A, D**). The temperature range is accordingly separated into three regions, glass (dark gray background), glass transition (light gray background), and SCL (white background).
**B**, **C**, **E**, **F** compare the relaxation time and shape exponent data resulting from conventional Kohlrausch-Williams-Watts (KWW) and KWW$_{MULTI}$ fitting of the $g_2$ data sets (each set consists of 96 mean values). The error bars represent the standard errors of the fits. As demonstrated in Fig. 1, KWW only provides meaningful results in the SCL state, while KWW$_{MULTI}$ excels in the glass and glass transition region but appears redundant in the SCL. Accordingly, the data sets are

reported only for the temperature regions where the different models provide reliable and more accurate fits (see also Supplementary Fig. 2 in the SI). The KWW fit parameters in the SCL show typical liquid behavior, namely a steep temperature dependence of $\tau$ and $\beta$ values below 1. In the glass transition region, the KWW$_{MULTI}$ approach reveals large differences between its two components KWW$_C$ and KWW$_S$: While the KWW$_C$ parameters follow glass-typical trends, in particular a relatively temperature-insensitive course of $\tau_C$ and a highly compressed shape, the KWW$_S$ parameters show liquid-like behavior in form of a steep temperature dependence of $\tau_S$ and a stretched shape.

The KWW$_C$ component of KWW$_{MULTI}$ combines non-equilibrium-associated properties like highly compressed $\beta_C$ values and a weak temperature dependence of $\tau_C$ in the whole glass and glass transition region, see Fig. 3. KWW$_C$ therefore exhibits the inverse behavior of KWW$_S$, as it thaws last during heating and freezes first upon cooling. Similar compressed decays in photon correlation spectroscopy studies were first reported for non-equilibrium materials like jammed colloids, concentrated emulsions, and clays[46,50,69–72]. Its appearance is mostly attributed to super-diffusive dynamics in form of ballistic motions, which manifest as collective, drift-like particle movements promoted by gradients of internal stresses that arise during jamming or vitrification[69,70,73]. Later XPCS studies confirmed compressed decorrelation to be also a general feature of vitrified metallic glass

formers[3,5,12,14,28–30], making a similar stress-based origin likely. Purely ballistic dynamics imply straight particle trajectories that create an archetypically compressed decay of the ISF with a $\beta$ value of 2[41,50,74–76]. Furthermore, they are characterized by a $|\mathbf{q}|$-dependent relaxation time according to $\tau \propto |\mathbf{q}|^{-1}$[69]. Recent XPCS results by Cornet et al.[14] suggest a rather ballistic-like $|\mathbf{q}|$-dependence also for glassy $Pt_{42.5}Cu_{27}Ni_{9.5}P_{21}$.

To state an interim conclusion, the heterogeneous and likely subdiffusive atomic motions of the α-relaxation process seem to survive, to some degree, in the non-equilibrium, as indicated by the presence of the typical stretched exponential decay (KWW$_S$). Yet, they appear superimposed by a second type of atomic motion that is characteristic for the non-equilibrium state and can be described by the compressed

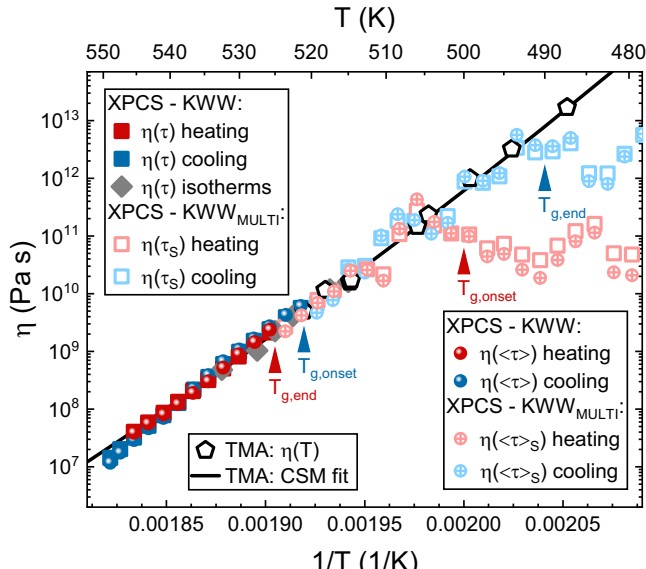

**Fig. 4 | Fragility comparison.** $\tau$, $\tau_S$, $<\tau>$, and $<\tau>_S$ data from Kohlrausch-Williams-Watts (KWW) and KWW$_{MULTI}$ fitting as well as $\tau$ data from isothermal X-ray photon correlation spectroscopy (XPCS)[13] are converted into viscosity values and compared to equilibrium viscosity data determined by thermomechanical analysis (TMA)[22], which serve as a reference. All data sets are described by the same cooperative shear model (CSM) fit curve, demonstrating that the different methods observe the same fragility. The start and end temperatures of the glass transition in the XPCS heating and cooling scans are indicated by red and blue arrows, respectively.

decay component (KWW$_C$). This latter process arises from longer range elastic interactions leading to ballistic-like particle motions, probably related to intermittent cluster dynamics[14,77,78]. The question arises, how these stress-induced ballistic-like motions can be visualized. The present homodyne experiment observes dynamics at the $|\vec{q}|$ value of the structure factor maximum and therefore probes both, the self and the distinct part of the ISF[42], thus reflecting the temporal decorrelation of the structure not only at the particle-particle distance but also on the MRO length scale[79]. Hence, specific traits of the stress fields that cause the compressed decay, such as the length and time-scale of the stress fluctuations, or the intensity of stress gradients, cannot be straightforwardly quantified from the current observations. Publications exploring the relation between the particle dynamics and the homodyne $g_2$ function in multicomponent systems in the non-equilibrium are still scarce[77], and therefore, we can only propose here a qualitative picture.

Nevertheless, many works interpret the glass transition in terms of heterogeneities in dynamics, structure, and density[44,45,80–86] and considering the earlier mentioned spatio-temporally heterogeneous dynamics imaged via ECS in a compositionally similar Pt-based alloy[65,66], such a scenario suggests itself in case of the present $Pt_{42.5}Cu_{27}Ni_{9.5}P_{21}$ system. We thus want to offer a picture that identifies spatial domains with slower relaxation times as the carriers of the stress-driven, ballistic-like motions. To illustrate our considerations in detail, we will focus on the cooling scan, where the SCL gently vitrifies without interfering aging effects as observed during heating. For this purpose, Fig. 5 connects the representative $g_2$ batches B4, B5, and B6 from Fig. 1 with simplified visualizations of the atomic dynamics at the given temperatures.

Figure 5A sketches the spatially heterogeneous dynamics of the equilibrium SCL. Here, the liquid-like and likely sub-diffusive motions of individual atoms are illustrated by orange arrows of varying lengths. Heterogeneity manifests spatially through nm-sized[65,66] domains of

diverse mobility, likely corresponding to fluctuations in density or atomic packing[84,87]. Domains with slower dynamics, which will be referred to as 'rigid' in the following, are highlighted by gray shadows to guide the eyes. In the SCL, the overall fast atomic mobility still enables the system to compensate perturbations immediately without 'falling out of equilibrium'. The measured $g_2$ decorrelation is of stretched exponential shape, indicating a broad distribution of relaxation timescales spanning over several decades, and can be fitted by a conventional KWW model.

With ongoing cooling, the dynamic heterogeneity further increases[65,66] while the overall free volume of the system decreases. Eventually, this must lead to the glass transition, which is a gradual, rate-dependent kinetic arrest[88,89] that manifests calorimetrically as a step in the system's heat capacity beginning at about 521 K, see Fig. 1A and Fig. 3D. It can be assumed that the more rigid domains play a decisive role in this process, as their slow dynamics likely lend them rather viscoelastic than liquid-like behavior[82], therefore enabling them to bear certain stresses in a solid-like manner. Hence, we pick up an approach firstly formulated by M. Cohen and G. Grest[86] and recently applied for polymeric melts[85] and interpret vitrification as an inter-locking and jamming process of the more rigid domains. The inter-locked domains then act as a gradually formed backbone structure that frustrates further volume shrinkage with ongoing cooling, and so, the system starts falling out of equilibrium. The volumetric frustration causes stress gradients among the rigid domains, forcing them to perform (small-scale) ballistic-like collective motions that could be imagined as drifts or rotations, as illustrated by the purple underlying arrows in Fig. 5B. A macroscale analogy can be seen in the dynamics of drift ice, where interactions within densely packed ice floe agglom-erations (representing the backbone structure formed by jammed rigid domains) create tectonic motions[90]. Eventually, the present homodyne XPCS experiment detects the velocity gradients[41] that go hand in hand with these ballistic-like 'microscopic drift' motions in form of the typical compressed decay modeled by KWW$_C$. Never-theless, the gradual nature of the glass transition implies that stress-relaxation also still occurs through liquid-like α-relaxation processes that remained partially active, and therefore, the KWW$_S$ parameters still follow their liquid-like trends. Accordingly, vitrification is char-acterized by rivaling types of atomic motions. This is mirrored in the ISF by different decorrelation components with roughly the same timescale, finally creating the typical cut-off appearance described by the KWW$_{MULTI}$ model.

At about 490 K, the end of the glass transition is reached, where the heat flow signal approaches the nearly constant glassy level, as shown in Fig. 1A and Fig. 3D. Here, the last portions of liquid-like α-relaxation processes become arrested and, therefore, $\tau_S$ and $\beta_S$ finally start departing from their equilibrium trendlines as seen in Fig. 3 and Fig. 4. Regarding our qualitative picture in Fig. 5C, the rigid dynamic backbone has broadly evolved, and stresses mainly relax through ballistic-like drift motions. Accordingly, the observed $g_2$ decorrelation is dominated by the typical compressed decay.

The vitrification in the cooling scan is a rather 'gentle' process[91–93]. The system leaves the equilibrium at a low fictive temperature thanks to the slow cooling rate. In contrast, the as-spun ribbon features a distinctly higher fictive temperature, since it has been vitrified with a rate in the order of $10^6$ K s$^{-1}$[33]. The resulting configurational differences between the as-spun and the slowly cooled glass are observed in the particle motion component corresponding to liquid-like dynamics, KWW$_S$, since $\tau_S$ and $\beta_S$ values significantly differ between the heating and the cooling scan (compare Fig. 3C, F). In view of our scenario, a higher fictive temperature configuration corresponds to a less rigid dynamic backbone and a faster liquid-like component. The higher atomic mobility in the as-spun glass allows for structural relaxation upon heating, as it is visible in Fig. 3B, C, where $\tau_S$ and $\beta_S$ age towards equilibrium. When these parameters have reestablished their

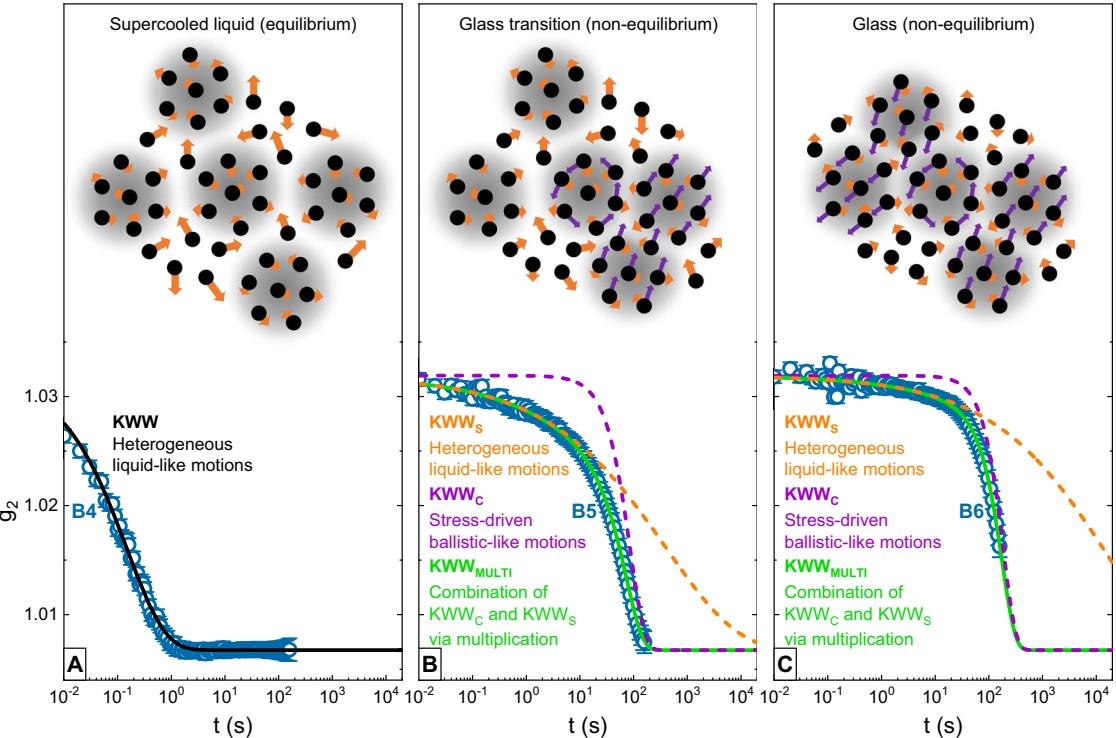

**Fig. 5 | Spatial heterogeneities, stress-driven dynamics, and the glass transition.** We present a possible scenario that illustrates the micro-scale dynamics during the cooling scan and connects it to the experimentally determined interplay of stretched and compressed decorrelation components. The shown $g_2$ curves consist of 96 mean value data points with standard deviations given by the error bars and correspond to the representative batches B4, B5, and B6 introduced in Fig. 1. Individual liquid-like atomic motions are symbolized by orange arrows. **A** In the equilibrium supercooled liquid state, spatio-temporally heterogeneous dynamics already have evolved. Regions with slower dynamics, termed rigid domains, are indicated by underlying gray shadows. The broad spectrum of relaxation times creates a stretched decay in $g_2$ that can be described by a conventional Kohlrausch-Williams-Watts (KWW) function. **B** Further cooling towards

vitrification leads to increased spatial heterogeneity as well as volume shrinkage of the system. Accordingly, the rigid domains start to gradually interlock and jam to form a rigid dynamic backbone. Further volumetric adaptations to temperature changes are frustrated, the system falls out of equilibrium and stress gradients arise. This causes the rigid domains to push, drift, and rotate against each other (underlying purple arrows), introducing collective ballistic-like atomic motions and a typical compressed decorrelation in $g_2$ that competes with the stretched liquid-like decorrelation component. Hence, the conventional KWW fit fails and the $KWW_{MULTI}$ approach is needed to model the complex shape of the $g_2$ decay. **C** In the glass, the rigid dynamic backbone is broadly established, and the compressed decay caused by the ballistic-like motions dominates the $g_2$ decorrelation. The fit parameters of the shown KWW and $KWW_{MULTI}$ curves can be found in Fig. 3.

equilibrium trendlines, further devitrification with increasing temperature basically occurs as the inverse vitrification process, namely via a gradual dissolving of the interlocked rigid regions until a full loss of compressed decorrelation characteristics is reached when approaching the equilibrium SCL.

In summary, we used slow temperature scanning XPCS to study the micro-scale dynamics of a Pt-based metallic glass former in the glass, glass transition, and SCL state upon heating and cooling. By assuming two different types of particle motion, we describe a scenario that reduces the difference between equilibrium and non-equilibrium again to a question of crossing timescales. The system remains in equilibrium as long as liquid-like atomic motions are able to relax perturbations within the timeframe determined by the underlying thermal protocol, creating a purely stretched decorrelation of the ISF. The glass transition upon cooling initiates as soon as rigid spatial domains start to interlock, causing volumetric frustration with further decreasing temperature. The resulting stress gradients induce collective particle movements in form of ballistic-like 'microscopic drift' motions, eventually creating a compressed $g_2$ decorrelation component that competes with the stretched decay component stemming from the liquid-like atomic motions. The here presented $KWW_{MULTI}$ approach allows us to model this interplay and clearly separate these coexisting decorrelation components. It shall be highlighted again that the fast exposure time of 0.01 s is the crucial technical condition to resolve the presence of such different

dynamic processes, as exemplarily shown in Fig. 2. Thus, we address the fact that superimposed stretched and compressed decay was never reported in earlier XPCS studies on metallic glasses to limitations of the temporal resolution. We expect that this superposition will become a regularly observed effect with future technological advances in XPCS, but possibly also for other experimental approaches like dynamic light scattering[40,51] or ECS[64–66]. Nevertheless, we are aware that the applicability of the $KWW_{MULTI}$ concept might be closely related to the probed length scales and experimental conditions of the present setup. Additional data at timescales at least two orders of magnitude faster than those probed here would definitely help to obtain a deeper understanding and might allow for a better discrimination between the additive and multiplicative fitting approaches. Finally, we are confident that temperature scanning XPCS will become a broadly used tool to study countless cases of non-equilibrium states[94–96] or transition effects in amorphous matter, e.g., regarding phase separations[97], liquid-liquid transitions[98], or secondary relaxations[99].

## Methods
### Sample preparation
The sample preparation of $Pt_{42.5}Cu_{27}Ni_{9.5}P_{21}$ consisted of inductively alloying the raw elements to obtain the master alloy, which was further purified through a fluxing treatment. An amorphous plate-shaped bulk specimen was produced from the master alloy by induction melting

and tilt-casting. For XPCS, amorphous ribbons with ~30 μm thickness were produced from this specimen by melt-spinning.

## XPCS experimental procedure

XPCS was carried out at the ID10 beamline at ESRF (after the EBS update). The measured as-spun ribbon piece was sanded to a thickness of 10 μm to achieve a regular cross section and an optimized signal-to-noise ratio. The partially coherent beam with a cross section of $10 \times 10$ μm² featured an energy of 21.0 keV and a flux of about $4.7 \times 10^{11}$ photons per second. The speckle patterns were obtained in frames of 0.01 s exposure time using an Eiger 4 M CdTe detector with a sample-detector-distance of 5100 mm. All measurements were performed at a fixed wave vector of $|\mathbf{q}| = 2.86$ Å⁻¹ ($\pm 0.05$ Å⁻¹), directly at the first sharp diffraction peak of the structure factor[13,100]. Supplementary Fig. 5 in the SI illustrates the setup. The as-spun ribbon sample was installed in a custom-build, PID-controlled furnace to apply a temperature scan program under high vacuum. In the first step, the sample was heated with a rate of 0.0167 K s⁻¹ from the glassy state into the SCL to 548 K. In the second step, the sample was subsequently cooled back into the glassy state, again with 0.0167 K s⁻¹. The TTCFs, $G(t_1, t_2)$, were calculated as

$$G(t_1, t_2) = \frac{\langle I(t_1)I(t_2)\rangle_p}{\langle I(t_1)\rangle_p \langle I(t_2)\rangle_p} \tag{8}$$

where $I(t_i)$ is the pixel intensity of the detector at the (absolute) experimental time $t_i$ and $\langle \ldots \rangle_p$ is the average over all pixels of the detector[24,26]. Considering that the intensity fluctuations are well described by Gaussian statistics and using the Siegert relationship from Eq. 2[101,102], the intensity autocorrelation functions, $g_2$, were obtained from averaging over the respective TTCFs. The calculation of TTCFs and $g_2$ data was performed in batches of 24000 frames, corresponding to a time span of $\Delta t = 240$ s or, respectively, a temperature span of $\Delta T = 4$ K. This batch size was found to be adequate for further KWW and KWW$_{\text{MULTI}}$ fitting as it provides enough temporal range to capture most of the decorrelation process while still retaining a reasonable error in temperature within each batch ($\pm 2$ K). To achieve a sliding average effect, consecutive batches overlapped halfway, hence by 12,000 frames (120 s, 2 K).

## Temperature correction procedure

To achieve a precise temperature calibration, further XPCS heating scans with the same experimental conditions as described earlier were performed on two other metallic glass forming systems, which show first signs of crystallization at about 390 K and 630 K, respectively. As soon as crystals appeared in the probed sample volume, they created quasi-stationary Bragg reflections in the speckle pattern. This leads to an abrupt and harsh increase of the $g_2$ baseline as shown in Supplementary Fig. 1 in the SI, allowing for a precise definition of the crystallization temperature $T_x$. DSC scans of these systems were performed with the same heating rate of 0.0167 K s⁻¹ using a Mettler Toledo DSC3 with Al crucibles under high-purity Ar flow. Here, $T_x$ was identified as starting point of the exothermal crystallization event in the heat flow signal, see also Supplementary Fig. 1 in the SI. Comparing the $T_x$ values defined by calorimetry and XPCS scans allowed to define a linear two-point temperature calibration that is applied for all XPCS data shown in this article. Supplementary Fig. 1C in the SI provides the correction formula.

## Ex-situ DSC scans

To provide heat flow signals for comparison and orientation, the temperature corrected heating-cooling program applied for the XPCS studies on the $Pt_{42.5}Cu_{27}Ni_{9.5}P_{21}$ ribbon was retraced ex-situ by DSC, again using the Mettler-Toledo DSC3 with Al crucibles under high-purity Ar flow. The as-spun $Pt_{42.5}Cu_{27}Ni_{9.5}P_{21}$ ribbons were heated from

463 K to 548 K and subsequently cooled back to the initial temperature of 463 K, using a rate of 0.0167 K s⁻¹. The resulting data has been evaluated and plotted using OriginPro 2021b.

## KWW and KWW$_{\text{MULTI}}$ fitting procedure

The first approach to fit the $g_2$ data using the KWW and KWW$_{\text{MULTI}}$ models, see Eq. 3 and Eq. 7, would be to leave all parameters free to obtain fitting curves that describe each individual data set to the best extent. Yet, at elevated temperatures in the SCL, the fast dynamics cause significant decorrelation even within the first time increment of 0.01 s, resulting in $g_2$ curves with too low initial heights, as demonstrated for example in Fig. 1C, D. Here, KWW fitting with free parameters leads to an underestimation of c. In contrast, the $g_2$ curves at low temperatures in the glassy state may not reach full decorrelation within the correlation window as can be seen in Fig. 1C, D, resulting in a misestimation of the baseline b in case of free parameterization. To solve these problems, we define fixed values for b and c, analogous to an XPCS analysis previously described in ref. 12. For b, an average value of 1.00675 is determined from those high-temperature batches that show full decorrelation. An average c value of 0.02517 is derived from low-temperature batches that show a full initial $g_2$ plateau. Hence, only $\tau$ and $\beta$ (or their respective counterparts from KWW$_{\text{MULTI}}$ fitting) remain as free fitting parameters. All the data fitting procedures have been performed using OriginPro 2021b.

Furthermore, the respective average relaxation times[58] $<\tau>$ and $<\tau>_S$ shown in Fig. 4 can be calculated according to

$$\langle \tau \rangle = \frac{\tau}{\beta} \Gamma\left(\frac{1}{\beta}\right) \tag{9}$$

## Aligning XPCS timescale data and equilibrium viscosity data

Equilibrium viscosities from TMA published in ref. 22 are shown in Fig. 4 and serve us as a reference in terms of the SCL state's fragility. To allow a direct comparison, the present $\tau$ and $\tau_S$ data from KWW and KWW$_{\text{MULTI}}$ fitting are transferred into the viscosity domain and aligned with the TMA data. To do so, the equilibrium viscosities are fitted in a first step using the cooperative shear model (CSM)[55,56]:

$$\eta(T) = \eta_0 \exp\left(\frac{T_g^*}{T}\ln\left(\frac{\eta_g}{\eta_0}\right)\exp\left(2n\left(1 - \frac{T}{T_g^*}\right)\right)\right) \tag{10}$$

Here, $\eta_0$ is the minimum viscosity reached at high temperatures of $4 \times 10^{-5}$ Pa s[22], $\eta_g$ is the viscosity value of $10^{12}$ Pa s commonly defined by convention as the glass transition value[1] and $T_g^*$ is the corresponding temperature value, which is 498 K in the present case. The only remaining free fit parameter n is a measure of the apparent fragility and is determined as $1.153 \pm 0.030$. This corresponds to an m-fragility of roughly 57[22,57] and a Vogel-Fulcher-Tammann (VFT) fragility parameter $D^*$ of 15.3[13,22,57]. In a second step, the CSM model is combined with the Maxwell relation[54], $\eta = G \tau$, to be applied in the timescale domain as

$$\tau(T) = \frac{\eta_0}{G}\exp\left(\frac{T_g^*}{T}\ln\left(\frac{\eta_g}{\eta_0}\right)\exp\left(2n\left(1 - \frac{T}{T_g^*}\right)\right)\right) \tag{11}$$

Fitting the equilibrium relaxation times (hence excluding data points that indicate vitrification due to deviation from the liquid behavior) from XPCS with this equation using the fixed $n = 1.153$ from the viscosity fit leaves only the shear modulus $G$ as a free fit parameter. The thereby determined $G$ values are $1.815 \times 10^8$ Pa ($\tau$, heating), $1.679 \times 10^8$ Pa ($\tau$, cooling), $6.05 \times 10^7$ Pa ($\tau_S$, heating), $2.66 \times 10^7$ Pa ($\tau_S$, cooling), $1.227 \times 10^8$ Pa ($<\tau>$, heating), $1.029 \times 10^8$ Pa ($<\tau>$, cooling), $2.02 \times 10^7$ Pa ($<\tau>_S$, heating), and $4.7 \times 10^6$ Pa ($<\tau>_S$, cooling). Now, the relaxation times can be

transformed into corresponding viscosity data by means of the Maxwell relation and each respective $G$ value. The resulting $\eta(\tau)$ and $\eta(\tau_S)$ values are depicted in Fig. 4. Their steepnesses agree well with the TMA equilibrium viscosities, indicating the same fragility among the data sets. Finally, it shall be stated that all fitting and data plotting of this study has been done using OriginPro 2021b.

## Reporting summary

Further information on research design is available in the Nature Portfolio Reporting Summary linked to this article.

## Data availability

The XPCS and DSC data generated in this study as well as the source files of all figures have been deposited in the figshare database under accession code https://doi.org/10.6084/m9.figshare.28855373.

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

## Acknowledgements

We acknowledge ESRF (Grenoble, France) for providing beamtime for the proposal HC-4479. We want to thank the technicians and staff at ESRF for their support in terms of the XPCS studies. We further thank Dr. Frank Aubertin for fruitful discussions. This project has received funding (in terms of A.C., J.S., B.R.) from the European Research Council (ERC) under the European Union's Horizon 2020 research and innovation programme (Grant Agreement No 948780). E.P. acknowledges financial support by grant PID2023-146623NB-I00 and the Maria de Maeztu Units of Excellence Programme grant CEX2023-001300-M funded by MICIU/AEI/10.13039/501100011033.

## Author contributions

Conceptualization: E.P., M.F. Methodology: E.P., N.N., M.F., Y.C., B.R. Validation: F.W., M.S. Formal analysis: E.P., M.F., N.N. Investigation: E.P., N.N., A.C., Y.C., F.Z., B.R., M.F. Resources: N.N., M.F. Visualization: M.F., E.P. Supervision: E.P. Writing—original draft: M.F., E.P., B.R. Writing—review & editing: N.N., S.S.R., A.C., Y.C., F.Z., L.R., B.A., M.N., F.Y., J.S., F.W., M.S., D.C., V.D.L., I.G., R.B.

## Funding

## Competing interests

The authors declare no competing interests.
