## [Transparent Peer Review file · Nature Communications]

Liquid-like versus stress-driven dynamics in a metallic glass former observed by temperature scanning X-ray photon correlation spectroscopy

Corresponding Author: Mr Maximilian Frey

Parts of this Peer Review File have been redacted as indicated to remove third-party material, and references to personal communication.

Version 0:

Reviewer comments:

Reviewer #1

(Remarks to the Author)

The review comments have been separately uploaded in a MS Word document

Reviewer #2

(Remarks to the Author)

The authors present a new thermal protocol in form of a continuous up- and down scan at constant rate, during which XPCS data was collected. The experimental procedure and the analysis are clear and well described, and the presented data is of significant interest to the community as it addresses the question after the origin of the complex dynamics observed in the glass transition range and below for metallic glasses. Overall, this appears to be an interesting and meaningful work of value to the metallic glass community.

Suggesting a factorization of two different types of dynamic processes, the data is interpreted as entailing a temperature-dependent process described by a stretched exponential decay dominating at $T > T_g$ and at the early stages of decorrelation at $T \approx T_g$ and below, and a second type of process of roughly constant timescale described by a compressed exponential function dominating the completion of decorrelation at $T \approx T_g$ and below. The authors suggest that the former is subject to a liquid-like type of motion that becomes visible thanks to the increased time-resolution while the latter is suggested to be due to a ballistic type of motion.

It is intriguing to see this convolution of two different types of dynamical processes characterized by stretched and compressed components. The observations made here seem to connect to previous XPCS-related findings, where experiments based on temperature-jumps or annealing at constant temperature result in either stretched (typical for $T > T_g$) or compressed (typical for $T \approx$ or $< T_g$) decorrelation behavior. This clearly identifies the foremost difference to previous experiments – the higher resolution in acquisition time and the more complex thermal protocol (temperature change at constant rate), resulting in the need for a factorization of two stretched/compressed exponential functions and thus the assumption of two dynamical processes to be able to fit the data in a representative way.

The comparability of timescale resulting from the stretched contribution at $T > T_g$ and at $T \approx T_g$ is convincingly shown in Figure 4. However, given the complex protocol, most pressing questions connect to this complexity and its influence on the here observed factorized convolution of dynamical processes.

Why is the factorized form of two dynamical processes not observed in less complex protocols such as temperature-steps or annealing at constant temperature? Shouldn't there be a thermal range at which this becomes visible even at acquisition times on the order of seconds? If this is a material intrinsic feature reflecting quenched-in stresses leading to ballistic-type

motion as described here, this behavior should also be observed in such situations. This apparently not the case, if fitting can be robustly achieved by a single stretched/compressed exponential function.

The compressed exponential part of the dynamics shows roughly constant fitting parameters with a stretching exponent beta of ca. 2 and time scales that are about 10 to 100 s (Figs 3B and 3E). Have the authors checked if this could be a small but still existent drift contribution from the setup itself that manifests itself at this time scale? As discussed for example in Gabriel et al., J. Chem. Phys. 142, 104902 (2015), drift is expected to manifest with an exponent of 2 – for higher temperatures the decorrelation of the material is fast enough, but at larger decorrelation time scales, i.e., at lower temperature, the decorrelation due to structural rearrangements would be slower and only significant during the early stages of the decorrelation, similar to what is observed here. A very convincing check is to measure dynamics under the here applied thermal protocol on material that not exhibit material-specific dynamics. In my opinion it is meaningful to include such a test in the here presented study, and it will strengthen the presented interpretation and discussion significantly.

The authors show six specific decorrelations (B1 to B6) out of a manifold of such data sets existent throughout the measurement. What do the two-time correlation functions for heating and cooling look like? It would be interesting to see the overall decorrelation behavior.

The authors mention that the factorization of eq. 5 is only applicable for distinctly different dynamical contributions, and that this is not fulfilled for decorrelation times beyond the initial three orders of magnitude in time of the decorrelations (mentioned in lines 190ff.). How robust is the outcome regarding fitting parameters? There are error bars on the data in Fig. 3, and also in the Supplementary Figure S2. Why is the strong deviation between single KWW fit and data at $T \approx$ or $< T_g$ not reflected in the error bar? I would have expected a much larger error bar on the solid squares in Figs. S2 B, C, E, F, but actually the error bars connected to the compressed functional form are much larger. Could the authors please explain why this is the case and what is implied in terms of robustness of the fits and interpretation of the results?

How does the averaging of the data (here over $\Delta t = 240$ s) effect the outcome of the fits in respect to the robustness of the factorization approach? If the window (i.e., Δt) is wider or more narrow, do the fitting results remain robust and comparable to the here presented outcome?

A question on some small details: What is the detector width, or the width of the q-range assumed as being “single q” at 2.86 AA?

Along the KWW fitting procedure, the values for the baseline b and the early plateau c are evaluated where accessible, and an average based on these values is fixed for data sets where this limits are not resolved. By how much do these values fluctuate? It might be interesting to see the evolution of these values with temperature and scanning direction.

Reviewer #3

(Remarks to the Author)

The paper uses powerful 4th generation synchrotron radiation to investigate the dynamics of a metallic glass former. The authors should be commended on their significant advances compared to previous synchrotron XPCS measurements; achieving measurement times of 10ms compared to seconds. However, the scientific impact is not at all clear. The equilibrium versus nonequilibrium arguments made are confusing and incorrect. The supercooled liquid is referred to an equilibrium state throughout the manuscript. Supercooled liquids are by definition metastable, as is the glassy state. Two different regimes of particle motion are assumed in modeling the supercooled liquid and glassy states. Why only two, could there be more than two? Heterogeneous liquid like atomic motions corresponding to the alpha-relaxation process have already been observed using XPCS. What is different here is the interpretation of the glassy state. Vitrification is described as the interlocking of nm-scale rigid domains with slower dynamics. But the suggestion of spatial heterogeneities and presence of medium range order in this metallic glass is not supported by any evidence and is tenuous at best. Most metallic glasses have little to no medium range order as indicated by the presence of a first sharp diffraction peak. Is there even a pre-peak in the structure factor of this glass at low-Q that could support your model?

Version 1:

Reviewer comments:

Reviewer #1

(Remarks to the Author)

The comments have been uploaded as a separate MS Word document.

Reviewer #2

(Remarks to the Author)

I do thank the authors for their detailed answers, which have indeed been very helpful and allowed me to better understand the presented work. I do recognize how thorough the authors' work is, and I appreciate that. However, I got some doubts based on an important aspect raised by reviewer 1, so that I do not support its publication in the present form.

As the authors very nicely and sincerely showed, fitting by the sum of contributing processes instead of their product also

yields convincingly converging fits. This is currently not at all communicated in the manuscript and sheds a different light onto the choice of the fitting function and thus the interpretation of the resulting parameters.

The authors base the discussion of a heterogeneous structure including the alpha process and a stress mode on the divergent evolution of dynamics. These are obtained by the multiplicative fitting approach only, while there are two approaches available that both fit the data convincingly. What proves that the proposed multiplicative fitting describes the data in a scientifically more reliable way? Why exactly should the multiplicative approach be preferred over the additive approach?

The basis for this decision-making together with the outcome from the alternative fitting approach needs to be made transparent in the manuscript.

Some arguments for this decision were given in the authors' response to reviewer 1. To me, however, the data suggests typical behavior at the glass transition, namely the freeze out of dynamical contributions given the error of the stretched-component time scales and is thus not an argument for excluding the obtained results as unphysical.

Only if these concerns are addressed convincingly, the interpretations of the findings as alpha and stress mode become meaningful.

Reviewer #3

(Remarks to the Author)

The authors have adequately addressed my concerns so I recommend publication. I still disagree with the blanket statement and the assertions made in references 46-48 that most metallic glasses explicitly have medium range order. Most do not exhibit a pre-peak and the observed periodicity simply arises from the packing of hard spheres, rather than some sort of preferred atomic ordering, see Price et al J. Phys. Condensed Matt. 1989 1 1005 and Salmon et al Nature 2005, 435, 75. I do agree however that this point is likely beyond the scope of this particular paper.

Version 2:

Reviewer comments:

Reviewer #1

(Remarks to the Author)

The authors have addressed all my comments. I am happy to recommend the manuscript for publication.

Reviewer #2

(Remarks to the Author)

First, a serious thank you for being so clear and helpful with your response to that major concern I raised.

Please let me give a detail response to the arguments you respond with, which are the following as I perceived them from your response letter, supplementary materials, and revised manuscript:

1) The additive approach yields $\tau_s \ll \tau_c$ by construction.

- As both timescales yield very comparable evolutions with temperature (essentially $\tau_s = \tau_c$ over the full temperature range), I also see the possibility that this additive approach "caps" the true evolution of τ_s , even though it is not known.

2) The additive model is mostly applied to the supercooled liquid regime to capture decoupling of alpha and beta mode.

- That is not a strong argument for or against the one or the other approach from my point of view.

3) The additive approach is slightly more scattered.

- The additive approach still captures the data convincingly and with reasonable error.

4) Based on the additive KWW-approach, the activation energy differs for heating and cooling. However, it differs strongly from what is expected based on mechanical relaxation experiments on comparable heating/cooling rates. The multiplicative KWW-approach yields a behavior for the stretched response part, that reflects the behavior from temperature-dependent mechanical relaxation experiments.

- I see this point and it struck me at first, as I think it is a very strong argument for why applying the multiplicative KWW-approach is crucial. I wonder why you do not make use of this argument in the manuscript. Currently, the manuscript refers to this point by stating that the activation energies based on the additive KWW-approach are "rather unphysical". However, also with this argument the important question remains: Why would one expect that in the XPCS data a de-convolution of structural relaxation dynamics is needed in order to be able to observe the dynamical behavior measured by mechanical relaxation experiments?

In the manuscript, the stretched dynamics was put in connection with heterogeneous liquid-like dynamics, while the compressed dynamical contribution connects to stress-driven motion. Is there a good reason why one would see the heterogeneous dynamics in mechanical experiments, but not the stress-driven part? Specifically, compressed dynamics have been observed in literature in mechanical relaxation experiments, such as in stress-relaxation experiments after small-strain excitation as in PRB 110134309 (2024), which should be referenced in the manuscript.

Concerns on my side have been addressed with care, which I appreciate. I am pleased that the authors discuss the choice of the multiplicative approach over the additive approach in the revised manuscript and included an additional section on the additive KWW-approach in the Supplementary Materials.

Apart from these aspects, I would like to mention some points that had been less pressing previously:

- 1) In the abstract, line 40-41, the authors write "This work not only extends the dynamical range probed by standard isothermal XPCS, but also clarifies the fate of the α -relaxation across the glass transition and provides a new perception on the anomalous, compressed temporal decay of the density-density correlation functions observed in metallic glasses and many out-of-equilibrium soft materials." I suggest changing "clarifies the fate" to "adds a different view".
- 2) I appreciate that the authors now include TTCFs in Figure 1. It would be helpful to insert color bars indicating the covered range or to make an according statement in the Figure caption.
- 3) In Figure 2, the orange and purple shading and arrows are very helpful, but the "KWWc" region is indicated very early if one looks at the corresponding contribution and the data. This "early onset" of the KWW_c distorts the perception within which temporal range the two different contributions dominate the dynamical response. Letting the KWW_c region start at 100 s would be more realistic instead of 10 s, and could be implemented easily.
- 4) In line 187, the authors write "KWWMULTI allows a way better description", which is quite subjective. I suggest changing this to "KWWMULTI allows for a more accurate fit of the data".
- 5) The newly added part (lines 291-304) starts with "It is important to stress that the presence of multiple dynamical contributions in the ISF can be modelled also with a sum of two KWW expressions (further termed KWWADD) ..." and would like to clarify the situation further by adding "with comparable accuracy" at the end of the here cited part.
- 6) Recently, outside the group of authors, XPCS-related work on the structural dynamics of metallic glasses in view of structural and dynamic heterogeneity (sub-diffusive transport, elastic backbone, stretched/compressed dynamics) were published, which are meaningful in context to the here presented results (Nature Communications (2024) 15:6595; Materials Today (2025) 82, 92-98) that should be referenced in the manuscript.
- 7) Reference 43 of the revised manuscript is not cited correctly in its current form.
- 8) In the SM, equation (S1) describes the transition coefficient as "f", while in Figure S7 it is referred to as "x". Also, in line 123 in the SM, it reads " $0 < f < 1$ " which should be " $0 < f <= 1$ ".

Rebuttal letter for

Liquid-like versus stress-driven dynamics in a metallic glass former observed by temperature scanning XPCS

First of all, we would like to thank all Reviewers for providing constructive criticisms that have directly improved the quality of our work.

Below we reply to each comment given by the reviewers. We hope that the following improvements can meet their expectations.

The text of the manuscript has been modified to take into account all the comments. In the revised manuscript and supplementary materials, we have highlighted changed parts in yellow.

Reviewer 1

Frey et al. presented an intriguing scientific story on the atomic scale dynamics transition during the glass-formation process of a Pt-based metallic glass former. In addition, the KWWmulti analysis method effectively decouples dynamics of different physical origins and can significantly benefit the XPCS community. I would like to invite the authors to consider the following suggestions that may further improve the quality of the manuscript:

We thank the reviewer for the positive comments and the thoughtful, in-depth suggestions to improve our work. In the following, we address all remarks point by point.

*(A) I believe the relaxation time scale of the system should include both the fitted time constant and the stretched exponential argument, as the latter can significantly change the shape of the function. Specifically, in H. Guo et al., PRL 109, 055901 (2012), $\tau = \tau_{SE} * \Gamma(\alpha-1)/\alpha$, where α is the stretch exponential argument and τ_{SE} is the time constant from the stretched exponential fit. The authors are encouraged to consider using this model for the calculation of viscosity in Fig. 4.*

We thank the referee for having raised this point which has been indeed largely debated among the authors.

In the case of a stretched shape, the relaxation function can be interpreted as a distribution of exponential relaxation modes $\phi(t) = \int_0^\infty D(\tau')e^{-t/\tau'}d\tau'$, and described by a KWW function $\phi(t) = e^{-(t/\tau)^\beta}$. The average relaxation time is then $\langle\tau\rangle = \int_0^\infty D(\tau')\tau'd\tau' = \int_0^\infty \phi(t)dt$, which for a KWW shape becomes $\langle\tau\rangle = \int_0^\infty e^{-(t/\tau)^\beta}dt = \frac{\tau}{\beta}\Gamma\left(\frac{1}{\beta}\right)$.

While we do agree that $\langle\tau\rangle$ as $\frac{\tau}{\beta}\Gamma\left(\frac{1}{\beta}\right)$ would be more appropriate for describing the density fluctuations in the supercooled liquid (stretched decay), we doubt on its meaning in the glass and glass transition regions, where the functional form of the total decay is not a single KWW function anymore, and multiple contributions influence the relaxation spectrum. In particular, the interpretation of $\langle\tau\rangle$ appears problematic regarding the compressed component that arises in the non-equilibrium, reflected in the paper by KWW_c. The reason behind this is that a compressed decay ($\beta>1$) cannot be explained through a possible distribution $D(\tau)$ of relaxation times. Therefore, the physical meaning of $\frac{\tau}{\beta}\Gamma\left(\frac{1}{\beta}\right)$ is not clear for a compressed relaxation.

For this reason, we have decided to not use $\langle\tau\rangle$ in the previous version of our manuscript. Nevertheless, we appreciate the referees remark and agree that $\langle\tau\rangle$ would be a valuable addition for the comparison of only the stretched KWW results in Fig. 4. Please see below and in the updated manuscript the new version of Fig. 4 that also includes $\langle\tau\rangle$ and $\langle\tau\rangle_s$, for which we find very similar behavior and temperature trends as for the KWW relaxation time counterparts, τ and τ_s .

Figure R1: Fig. 4 from the manuscript, but with additionally with aligned $\eta(\langle\tau\rangle)$ and $\eta(\langle\tau\rangle_s)$ data.

In the same vein, the authors pointed out that the improved fitting quality in Fig. 2 and 3 is a direct outcome of increasing the number of fit parameters, which I very much agree. Given that β does not vary significantly over the entire temperature range especially for the stretched KWW, the authors are encouraged to re-fit the data with fixed β to improve the confidence of the fitting.

We tried KWW_{MULTI} fitting with both β values fixed (e.g. $\beta_c=2$ and $\beta_s=0.3$), but in this case, the fits mostly do not converge. Since we use fixed values for the baseline, b , and the contrast, c (see the ‘Materials and Methods’ section of the article), fixing both β values seems to erase too many degrees of freedom to allow for a robust fitting procedure. Nevertheless, we note that β_s ranges close to 0.3, at least in case of the cooling scan. Hence, fixing only $\beta_s=0.3$ allows for stable fitting, please find the results below in Fig. R2. Besides some offsets in τ_s and β_c , the

results are comparable to those shown in the manuscript. Fixing β_S at higher values similar to those known from the KWW fitting in the SCL state, let's say at a value of about 0.5, results in dysfunctional fitting which cannot accommodate the g_2 shape. We note that the low value of β_S agrees well with the random first order transition theory (RFTO) which predicts a decrease of β on entering the glass state until a vanishing value at the Kauzmann temperature ¹.

Figure R2: Comparison KWW_{MULTI} fitting results of the cooling scan g_2 data with free shape parameters (as shown in the article) and with fixed $\beta_S=0.3$.

As the different fitting procedures lead to similar results, we have preferred to keep the unfixed β_S parameter, as it describes more accurately the results and in particular the weak temperature trend in β and β_S reflecting the increasing dynamic heterogeneity with increasing undercooling as discussed in the article and in agreement with other published data ².

In addition, it is highly likely that a linear superposition of multiple single exponentials, or one stretched exponential and one compressed exponential, could yield similar fitting quality compared to the KWWmulti model in the manuscript. Have the authors considered this alternative approach and how it may alter the scientific story?

The referee raised an important point. The sum of multiple exponentials was discarded, as it cannot reproduce an overall compressed shape. Any distribution of single exponentials gives always rise to a stretched shape or, if the timescales are well separated, to multiple step decays. On the other hand, the sum of one stretched and one compressed relaxation can indeed adapt to the observed experimental shape. The possibility of an additive g_2 fit combining one stretched and one compressed formula is indeed very briefly mentioned in the original manuscript:

“If spatial regions of the probed volume become extremely different in dynamics, with well separated timescale distributions, the heterogenous dynamics evolve from a stretched to a step-like decay. Such clearly separated relaxation phenomena can be modelled by a sum of two (or more) KWW expressions as given in Eq. 1³⁻⁶.”

For example, we tested an additive g_2 fit model of the form

$$g_2(t) = b + c \left[f \exp\left(-\left(\frac{t}{\tau_s}\right)^{\beta_s}\right) + (1 - f) \exp\left(-\left(\frac{t}{\tau_c}\right)^{\beta_c}\right) \right]^2. \quad (\text{R1})$$

This formalism implies a gradual crossover on cooling a supercooled liquid, with the $g_2(t)$ evolving from a liquid-like stretched decay to a compressed shape on tuning the parameter f that could be termed ‘transition coefficient’, with $0 \leq f \leq 1$. $f=1$ thereby indicates a sample that is completely equilibrated in the SCL state and $f=0$ means that a fully vitrified system. This model has its undeniable elegance, since it allows a fluent transition from stretched to compressed decay during vitrification, which might appear intuitive at a first glance and it has been used in many previous studies³⁻⁶. An example of this fitting the g_2 data is shown in Fig. R3.

Figure R3: A g_2 curve obtained upon heating. The temperature is located at the onset of the glass transition event. The red fit curve is obtained with the additive model from equation (1) and describes the data set well.

The additive and the multiplicative models are compared in Fig. R4. The transition coefficient f roughly follows the expected trend, featuring values around 1 in the SCL and changing towards 0 when the glassy state is approached. However, the τ and β values produced by the additive model show more scatter, which we were not able to change by any means of ‘fitting tricks’ (e.g. fixing f according to the sigmoidal occurrence of the calorimetric glass transition). More importantly, the additive model shows fast dynamics for the stretched component below T_g as by construction locates the stretched component always at nearly equal or shorter time scales than the compressed contribution. In this scenario, the α -process would convert in a fast localized process at the probed length scale, followed by the stress mode. This interpretation contrasts with the general assumption of an increasingly slower structural relaxation process, and therefore we have discarded it. We would like to note that the majority of the works reported in literature employ the additive model to describe the decoupling between the slow α -relaxation process with respect to faster localized processes in the supercooled phase and are therefore representing a different part of the relaxation spectrum with respect to the slow out-of-equilibrium dynamics reported in this work.

Our multiplicative approach is somewhat similar to the one in reference ¹, in which the authors consider two processes taking place simultaneously for all particles, leading to a convolution of the scattering functions in the frequency domain $S(q,\omega)$ but to a simpler multiplication in the

time domain $F(q,t)$. We have introduced citations to these ref. ^{7,82} in the new text. In these references the authors are considering a fast process in the liquid which is of different nature than the compressed component in our work, but we think they illustrate the physical meaning of considering a multiplicative model for the $F(q,t)$.

Figure R4: Comparison of fitting results of the multiplicative approach from the main text (KWW_{MULTI}) with those from the additive approach introduced in Eq. R1.

(B) Regarding the flow of the paper: I would like to invite the authors to consider the possibility of moving Fig. 4 from the Discussion section to the Result section, and lead the Discussion section with the physical picture in Fig. 5. This will hopefully highlight the interesting physics and make the manuscript more stream-lined and easier to absorb. This flow should also help filter out some of the text in the Discussion section which are not directly relevant to Fig. 5.

We thank the referee for this very nice suggestion, and we changed the manuscript accordingly. Indeed, the rather ‘technical’ considerations around Fig. 4 fit way better to the results section and the discussion section becomes more focused and more straightforward.

(C) Regarding the summary paragraph: have the authors considered elaborating the significance of KWWmulti in decoupling dynamics in complicated solid-state systems? As the authors pointed out, the compressed exponentials cannot be expanded in stretched exponentials and vice versa. Consequentially, the fitting is unique within the KWWmulti framework, and each component represents a dynamic mode with a distinct physical origin. More emphasis on the technical impact and the scientific perspectives, especially in other systems or related topics, will help end the paper on a stronger note.

The appearance of compressed decays is seen in several systems, in many cases linked to changes in dynamical regimes or far from equilibrium conditions, as in ⁹⁻¹¹. Our work may provide a way of interpreting some of these results and provide new important insights on the glass transition in a region usually inaccessible by many simulation and experimental techniques. At the same time, our results raise also several questions on the nature of the stress-mode that will require further studies in a broader time and length scale range in order to develop a proper picture of the glass dynamics and the interplay between the two processes. Please find the summary paragraph now with added references.

Other technical comments:

1. The abstract mentioned the use of modern detector and 4th generation x-ray source. The CdTe detector is indeed relatively novel but the use of high-energy x-ray beam is not fully justified in the manuscript. In addition, it is not clear that the data was collected at ESRF or ESRF-EBS. The authors are recommended to provide further clarification.

We apologize for not having been enough clear in the previous version of the manuscript. The measurements were conducted at ESRF after the EBS update, profiting of the larger coherent flux available at the ID10 beamline, which allows to perform our temperature scanning XPCS. The beamtime took place in July 2021 and was one of the first experiments at ID10 after the upgrade. Due to the Covid crisis, all beamlines have suffered a delay in the delivery of new optical components. At that time, the available detectors and focusing system were better adapted to high energy configurations and this is the reason why we worked at 21 keV. In addition, with the EBS the coherent intensity at 21 keV was increased by 75 times.

2. The authors claimed that “This non-isothermal approach can only be sparsely found in literature, and we apply it for the first time on a metallic glass former, ...”. Temperature-varying in-situ measurements are quite common in XPCS, e.g., PRL 100, 055702 (2008) and PRL 109, 165701 (2012), or in soft materials such as PRL 119, 178006 (2017) and J. Chem. Phys. 151, 104902 (2019). The authors are recommended to rephrase this statement.

We agree that this section lacks some clarity. We adapted the text to highlight that we refer to a temperature scanning method that uses a constant temperature change and refer to some previous similar approaches. The updated version of the manuscript now features the following text:

“XPCS is usually measured under isothermal conditions. In the present study, we instead perform temperature scanning XPCS, where the temperature changes continuously with a constant rate. In order to do so, we use the high flux available at the ID10 beamline of the fourth-generation synchrotron radiation facility ESRF after the EBS upgrade^{12,13}. Such continuous temperature scanning XPCS approaches can be found in literature^{5,14-16}, yet we apply it for the first time on a metallic glass former, namely the Pt_{42.5}Cu₂₇Ni_{9.5}P₂₁ alloy¹⁷.”

3. None of the g_2 in the paper has error bars. In addition, the Method section included the calculation for the two-time correlation function, however no two-time correlation function is shown in the manuscript or SI. Given the physical picture in Fig. 5, it is likely that heterodyning can be observed in two-time correlation with sufficient statistics. The authors are encouraged to provide error bars for g_2 and showcasing some binned two-time correlation function.

You are right, the error bars are missing. We changed Figs. 1, 2, and 5 accordingly.

Regarding the two-time correlation functions, we have included in the updated Fig. 1 six representative two-time correlation functions corresponding to temperature intervals of 4 K (240 s) centered at three temperatures, both during the heating and cooling ramps (batches B1-B6). See below and in the accordingly updated manuscript.

Inspecting the two-time correlations and the g_2 tails, we do not observe detectable signatures of heterodyning. The details of the ‘microscopic atomic drift’ that we attribute to the compressed process and that we sketch in Fig. 5, for instance what is its characteristic length scale, are not possible to unveil at this stage. In Fig. 5 we propose a kind of ‘disordered’ drift, comprising

gradients of velocity with length scales of the order of nanometers, which may come from a process of heterogeneous densification and the corresponding accumulation of internal stresses, but the depicted length scale is just a guess. The only thing we can assert is that the $g_2(q,t)$, at q corresponding to the structure factor maximum does not show any indication of heterodyning.

Figure R5: Fig. 1 from the updated manuscript. Here, TTCFs are included and error bars for the g_2 data are added.

4. The authors claimed that “While the signal-to-noise ratio and detector technology limited the exposure time to several seconds in earlier studies, recent advances allowed exposure times in the sub-second regime”. Given that the kHz-frame rate detector is the current industrial standard, “Recent advances” most likely means 4th-generation sources like ESRF-EBS, however it is not clear whether the data was taken with ESRF-EBS. The authors are encouraged to rephrase their statement.

We have modified the text to explain this part better. As written above, the measurements were performed after the EBS upgrade which allows to have enough coherent flux for temperature scanning XPCS at the atomic scale.

5. The manuscript does not contain any detector images of the scattering feature. The detector images contain important information such as the region-of-interest (ROI) definition and the scattering rate in terms of count-per-pixel. In addition, the onset of the crystal peaks during the nanodomain formation as the authors pointed out in the manuscript would certainly be of interest to the community. The authors are recommended to provide representative scattering patterns at different stages of the glass formation and specify the ROI.

We believe that the referee has misunderstood our results as we do not claim the formation of crystalline nanodomains but of amorphous rigid regions. The experimental design of WAXS-XPCS analyzes the speckle patterns in a very small region of the reciprocal space, at the maximum of the structure factor (see an example in Fig. R6). The q range captured by the detector is 0.3 \AA^{-1} . The area of the detector was divided in three ROI, each of one corresponding to a width q of 0.1 \AA^{-1} . The middle ROI containing $q_{\text{max}}=2.86 \text{ \AA}^{-1} (\pm 0.05 \text{ \AA}^{-1})$ was selected for the analysis. In the materials and methods section, the information is accordingly added, please see the highlighted text. During the experiments, we monitored the evolution of the structure by measuring the corresponding static structure factor in the $0.5\text{-}3\text{\AA}^{-1}$ wave vector range, without observing any sign of crystallization.

Figure R6: An example of a diffraction pattern of a metallic glass former. The ROI used for XPCS is located at the center of the FSDP. Here, the respective speckle pattern is rather uniform and featureless, as shown in the inset.

6. Fig. 1 shows that not all g_2 decay from the same point, which may be a sign for faster dynamics. Is the contrast factor f fixed for all the fittings presented in Fig. 2 and 3? This should be stated in the manuscript along with the justification as it can alter the fitting results.

Exactly, at elevated temperatures in the SCL, the observed contrast drops since considerable parts of the decorrelation already occur within the initial acquisition frame of 0.01 s. On the other hand, decorrelation does not occur completely at low temperatures. The technical aspects of the fixing procedure are not mentioned in the results or discussion section, but are included in the ‘Materials and Methods’ section, which we quote here:

“KWW and KWW_{MULTI} fitting procedure

The first approach to fit the g_2 data using the KWW and KWW_{MULTI} models, see Eq. 3 and Eq. 6, would be to leave all parameters free to obtain fitting curves that describe each individual data set to the best extent. Yet, at elevated temperatures in the SCL, the fast dynamics cause significant decorrelation even within the first time increment of 0.01 s, resulting in g_2 curves with too low initial heights, as demonstrated for example in Fig. 1C and D. Here, KWW fitting with free parameters leads to an underestimation of c . In contrast, the g_2 curves at low temperatures in the glassy state may not reach full decorrelation within the correlation window as can be seen in Fig. 1C and D, resulting in a misestimation of the baseline b in case of free parameterization. To solve these problems, we define fixed values for b and c , analogous to an XPCS analysis previously described in ¹⁸. For b , an average value of 1.00675 is determined from those high-temperature batches that show full decorrelation. An average c value of 0.02517 is derived from low-temperature batches that show a full initial g_2 plateau. Hence, only τ and β (or their respective counterparts from KWW_{MULTI} fitting) remain as free fitting parameters.”

We found that this fixing procedure significantly increases fit stability and consistency of the resulting data but does not alter the fitting results fundamentally (hence, the fixing approach does not ‘create’ new effects but improves the data quality). If you are further interested in the determination procedure used to define b and c , please feel free to take a look at the answer to the last question of reviewer 2 (page 24).

7. In the discussion of the origin of the stretched exponential, it was mentioned that the stretched exponential arises from the polydispersity of the time constant, “with a distribution $D(\tau)$ stretching over several decades”. However, simply combining two dynamic modes with time

constants that are orders of magnitude apart will most likely result in a double-stage decay. Could the authors provide clarification on the origin of the stretched exponential and perhaps add more references?

We apologize for the confusion. The explanation of the presence of a relaxation time distribution lacks some references. We added some review articles by Ediger and Richert^{19–21} in the respective part of the results section.

A stretched exponential decay, as the one expected for the α -relaxation in liquids²², can be interpreted as the participation of a broad distribution of exponential relaxation modes. The assumption of only two exponential modes is a simplified version of this dynamic heterogeneity. Participation of only two exponential modes leads certainly to a stretching of the overall function but it cannot reproduce the whole KWW shape with a β near 0.5, which derives from a relaxation spectrum that contains a long high frequency tail expanding several decades in timescale.

While the explanation in the results section is rather short, we discuss the nature of spatio-temporal heterogeneous dynamics to a greater extent in the first segment of the discussion section (for us, it appears to be most conveniently placed in the discussion), which we quote here:

“The shape exponents β and β_s in Fig. 3C and F feature values distinctly below unity that reflect the stretched exponential decay typical for the heterogeneous nature of supercooled metallic glass forming liquids^{2,18,23,24}. Furthermore, β and β_s decrease with decreasing temperature, as previously observed in the isothermal XPCS studies by N. Neuber et al.². This temperature trend can be attributed to the fact that liquid dynamics become more temporally^{25–28} and spatially^{20,21,27} heterogeneous with undercooling. While the former describes a general, non-localized tendency towards a broadening relaxation time distribution, $D(\tau')$ (see Eq. 5), the latter specifically refers to spatial fluctuations in the dynamics^{21,29}. P. Voyles et al. recently used novel electron correlation spectroscopy (ECS)³⁰ to image such spatio-temporally heterogeneous dynamics in supercooled $\text{Pt}_{57.5}\text{Cu}_{14.7}\text{Ni}_{5.3}\text{P}_{22.5}$, an alloy similar to our present system^{31,32}. Large differences in relaxation time of up to two orders of magnitude were observed between neighboring nm-sized domains, thus, on a length scale that is typical for the medium range order (MRO). This implies a sub-diffusive^{25,33,34} structural relaxation process that is governed by cooperative atomic rearrangements and caging effects.”

We try to solve the issue by listing the term $g_1(t) = \int g_{1,\tau'}(t)D(\tau')d\tau'$ as an actual equation with a number (Eq. 5 in the updated manuscript), which we then cite in the discussion section to establish a direct connection.

8. Fig. 2 discusses different regions where the stretched and the compressed dynamics are dominating. Is it possible to label these regions on the figure?

Thank you for your constructive suggestion. We increased the clarity of Fig. 2 by indicating the regions dominated by stretched and compressed decay via colored backgrounds and arrows, please see below in Fig. R7 and in the updated manuscript.

Figure R7: Updated version of Fig. 2 of the main article. Colored backgrounds and arrows help to identify the dominant regions.

9. In Fig. 1 and 3, the (A), (B), (C) labels are blocking the data points. Replotting is recommended.

Thanks for the hint, we modified the figures accordingly.

10. In Fig. 5, the suppression of KWWs is not very clear from the illustrative drawing. Maybe the orange arrows in the supercooled state can be drawn a bit longer?

We significantly increased the arrow size in the SCL and glass transition region. Now, the temperature-dependent dynamical changes should be clearer, thanks for this suggestion.

11. What is the unit of '1/s' in XPCS Methods?

We apologize for the unclear formulation; we were meaning the number of photons per second. We corrected it.

Reviewer 2

The authors present a new thermal protocol in form of a continuous up- and down scan at constant rate, during which XPCS data was collected. The experimental procedure and the analysis are clear and well described, and the presented data is of significant interest to the community as it addresses the question after the origin of the complex dynamics observed in the glass transition range and below for metallic glasses. Overall, this appears to be an interesting and meaningful work of value to the metallic glass community.

Suggesting a factorization of two different types of dynamic processes, the data is interpreted as entailing a temperature-dependent process described by a stretched exponential decay dominating at $T > T_g$ and at the early stages of decorrelation at $T \approx T_g$ and below, and a second type of process of roughly constant timescale described by a compressed exponential function dominating the completion of decorrelation at $T \approx T_g$ and below. The authors suggest that the former is subject to a liquid-like type of motion that becomes visible thanks to the increased time-resolution while the latter is suggested to be due to a ballistic type of motion.

It is intriguing to see this convolution of two different types of dynamical processes characterized by stretched and compressed components. The observations made here seem to connect to previous XPCS-related findings, where experiments based on temperature-jumps or annealing at constant temperature result in either stretched (typical for $T > T_g$) or compressed (typical for $T \approx$ or $< T_g$) decorrelation behavior. This clearly identifies the foremost difference to previous experiments – the higher resolution in acquisition time and the more complex

thermal protocol (temperature change at constant rate), resulting in the need for a factorization of two stretched/compressed exponential functions and thus the assumption of two dynamical processes to be able to fit the data in a representative way.

The comparability of timescale resulting from the stretched contribution at $T > T_g$ and at $T \approx T_g$ is convincingly shown in Figure 4. However, given the complex protocol, most pressing questions connect to this complexity and its influence on the here observed factorized convolution of dynamical processes.

We thank the referee for the positive assessment and the kind words. We appreciate the profound examination of our work. We will address all the remarks in the following.

Why is the factorized form of two dynamical processes not observed in less complex protocols such as temperature-steps or annealing at constant temperature? Shouldn't there be a thermal range at which this becomes visible even at acquisition times on the order of seconds? If this is a material intrinsic feature reflecting quenched-in stresses leading to ballistic-type motion as described here, this behavior should also be observed in such situations. This apparently not the case, if fitting can be robustly achieved by a single stretched/compressed exponential function.

This is a very good point. To quickly answer: it can be observed. The results are of course independent on the applied thermal protocol as intrinsic of the glass state. The reason why it was not observed in previous studies is the limited accessible time window in pre-ESRF-EBS studies.

As example, Fig. R8 shows unpublished isothermal data, now regarding a Pt-Pd-based ($\text{Pt}_{30}\text{Pd}_{12.5}\text{Cu}_{27}\text{Ni}_{9.5}\text{P}_{21}$) system similar to the one investigated in the manuscript ($\text{Pt}_{42.5}\text{Cu}_{27}\text{Ni}_{9.5}\text{P}_{21}$). The measurements were performed at ID10 beamline at ESRF with acquisition times of 0.1 s and 0.01 s. The data show a multicomponent decay as those reported in our article. The curves cannot be described by a normal KWW function, confirming the validity of our approach with the $\text{KWW}_{\text{MULTI}}$ fitting.

Hence, we think these examples support the idea that the cut-off appearance in the g_2 data could be a general feature of metallic glass formers in the non-equilibrium state and that the multiplicative KWW model seems to be the adequate approach to describe it. Nevertheless, the quantity of future XPCS studies will provide the ultimate verification for the model.

Figure R8: Isothermal sub- T_g XPCS studies on a Pt-Pd-based system. KWW fails, but KWW_{MULTI} can describe the non-equilibrium decorrelations.

The compressed exponential part of the dynamics shows roughly constant fitting parameters with a stretching exponent beta of ca. 2 and time scales that are about 10 to 100 s (Figs 3B and 3E). Have the authors checked if this could be a small but still existent drift contribution from the setup itself that manifests itself at this time scale? As discussed for example in Gabriel et al., J. Chem. Phys. 142, 104902 (2015), drift is expected to manifest with an exponent of 2 – for higher temperatures the decorrelation of the material is fast enough, but at larger decorrelation time scales, i.e., at lower temperature, the decorrelation due to structural rearrangements would be slower and only significant during the early stages of the decorrelation, similar to what is observed here. A very convincing check is to measure dynamics under the here applied thermal protocol on material that not exhibit material-specific dynamics. In my opinion it is meaningful to include such a test in the here presented study, and it will strengthen the presented interpretation and discussion significantly.

Thank you for this important remark. Transit decorrelation is indeed a very important aspect for our measurement setup, since changing temperature goes hand in hand with thermal expansion of the oven and sample holder, which creates a sample displacement in relation to the incoming X-ray beam. In other words, the continuous heating and cooling causes a slight constant movement, a drift of the sample over time, and this effect must be considered to exclude measurement artifacts. Please allow us to explain the details in the following.

The article by S. Busch et al.³⁵ which we cite in the manuscript also explains the influence of transit decorrelation in detail and addresses it to a purely compressed decorrelation footprint.

[REDACTED]

Hence,

you will find his respective publication³⁶ already cited in the original version of our manuscript.

Taking the transit decorrelation into account expands Eq. (5) from the manuscript to $|g_1(t)|^2 = |g_{1,S}(t)|^2 |g_{1,C}(t)|^2 |g_{1,Transit}(t)|^2$, as schematically illustrated in Fig. R9.

Figure R9: Schematic illustration of a temperature scanning XPCS measurement on a non-equilibrium metallic glass sample and the rivalry between KWWs, KWW_C, and the transit decorrelation. The ribbon sample experiences a thermal drift (gray arrow) due to the underlying constant temperature change and the resulting thermal expansion of the setup (perpendicular to the incoming X-ray beam).

With the transit decay issue in mind, we chose the used rate of temperature change (1 K/min) carefully and by approximating the transit decorrelation based on the geometrical conditions of the measurement setup. These considerations were originally part of the manuscript but were erased in a ‘streamlining attempt’ in order to focus on the main message and to create a more readable manuscript.

But in view of all the thoughtful and in-depth reviewer remarks, we are happy to reintroduce such details. In results section of the article, you will now find the following remarks:

“For the sake of completeness, we want to note that a compressed decay in g_2 can also stem, in principle, from a macroscopic sample movement in relation to the incoming photon beam,

called transit decorrelation^{35,36}. Indeed, the constant temperature change of the temperature scanning method implies a corresponding thermal expansion of the measurement setup, which could cause such an artifact. Yet, we estimated this effect based on the conditions of the given experimental setup and found it to be negligible, as further explained in the supplementary information.”

The added explanations in the supplementary information are as follows:

“*Macroscale transit decorrelation*

In the present experimental setup, oven and sample holder are arranged perpendicular to the incoming beam. Therefore, the continuous temperature change provokes thermal expansion that causes ‘refilling’ of the scattering volume through sample transit with a drift velocity v calculated as

$$v = L \alpha \frac{dT}{dt} \quad (\text{B})$$

with a temperature-affected setup length of approximately $L=7$ cm and $\alpha=13$ $\mu\text{m}/(\text{m K})$ being the thermal expansion coefficient of Nickel (the material the oven is made of). Assuming a Gaussian shape of the beam’s intensity profile, the resulting decorrelation in $|g_{1,Transit}(t)|^2$ can be mathematically described³⁵ as

$$|g_{1,Transit}(t)|^2 \propto \exp\left(-\left(\frac{t}{\tau_{Transit}}\right)^2\right) \quad (\text{C})$$

where $\tau_{Transit}$ is the timescale of transit decorrelation that is calculated from v and the width of the beam spot $h=10$ μm as

$$\tau_{Transit} = \frac{h}{v}. \quad (\text{D})$$

In case of the applied heating rate of 0.0167 K/s, $\tau_{Transit}$ can be estimated as 659 s. Fig. S4 shows the same data and fit curves as Fig. 1 in the main article, only adding the transit decorrelation (dashed line). Due to the shape exponent of 2, the transit decorrelation is highly compressed, and the vast majority of the decay is observed outside of the observation window of 240 s. Hence, a significant influence of transit decorrelation on the present measurements can be excluded.”

Also, you can find an explanatory figure in the supplementary information (Fig. S4), which is also shown here in Fig. R10.

Figure R10: The g_2 curves of the representative batches B1-B6, together with the estimated decorrelation signal due to sample transit that stems from the continuous thermal expansion of the experimental setup, see the gray dotted line. This decorrelation mechanism shows no significant impact within the applied observation window of 240 s.

We agree that measuring a reference material without any material-specific dynamics with the same thermal program would be a nice approach. We remind the referee that additional data would require the submission of a new proposal in a very highly competitive beamline, with to the best of our estimation, the possibility to acquire data after next summer, being the next proposal deadline in March 2025, and thus delaying the article of more than one year for a confirmation that we believe is already provided by the above discussions, especially regarding the KWW_{MULTI} behavior in the shown isothermal XPCS data, where a compressed decorrelation due to thermal drift can be excluded.

The authors show six specific decorrelations (B1 to B6) out of a manifold of such data sets existent throughout the measurement. What do the two-time correlation functions for heating and cooling look like? It would be interesting to see the overall decorrelation behavior.

Thank you. Your remark is similar to the one of reviewer 1, therefore we would like to refer to the previous answer, see the respective text in the context of Fig. R5. As mentioned there, we updated the Fig. 1 in the manuscript with included TTCFs for the six batches B1 to B6. We hope this addresses the issue and enhances the clarity and quality of the work.

The authors mention that the factorization of eq. 5 is only applicable for distinctly different dynamical contributions, and that this is not fulfilled for decorrelation times beyond the initial three orders of magnitude in time of the decorrelations (mentioned in lines 190ff.). How robust is the outcome regarding fitting parameters? There are error bars on the data in Fig. 3, and also in the Supplementary Figure S2. Why is the strong deviation between single KWW fit and data at $T \approx$ or $< T_g$ not reflected in the error bar? I would have expected a much larger error bar on the solid squares in Figs. S2 B, C, E, F, but actually the error bars connected to the compressed functional form are much larger. Could the authors please explain why this is the case and what is implied in terms of robustness of the fits and interpretation of the results?

We think that the difference in error bar size between the KWW results and the KWW_{MULTI} results (especially regarding β_c) must be seen in the light of the number of free fitting parameters (degrees of freedom) and the amount of g_2 data points that contribute to the fitting. KWW fitting only features two free parameters (β and τ), since c and b are fixed (please see the respective discussions about the fixing procedure). KWW_{MULTI} instead features four free fit parameters. As discussed in the manuscript and illustrated in Fig. 2 (or in Fig. R7 here in the rebuttal), the fits for KWW_S and KWW_C are dominated by different portions of the g_2 set: while KWW_S is predominantly defined through the initial part of the data, KWW_C is predominantly determined by the latter part, where the cut-off appearance is visible. Hence, KWW_S and KWW_C effectively result from reduced data set sizes in comparison to KWW, which is determined by the whole data set. Hence, the error bars tend to increase for the KWW_{MULTI} results.

Nevertheless, KWW_{MULTI} provides obviously better fitting quality than KWW when we look for the non-equilibrium state. This is directly quantified by the coefficient of determination (R^2). For example, the fits of batch B5 (glass transition during cooling, please see Fig. 1) feature an R^2 of 0.96747 for KWW and an R^2 of 0.99883 for KWW_{MULTI}.

Regarding the robustness of fitting, we can say that we used batch processing to fit all g_2 curves of one temperature scan (heating or cooling) at once. The most important aspect here is to implement meaningful start values for the β parameters, e.g. 0.5 and 1.5 for β_s and β_c . This way, the KWW_s and KWW_c components adapt nicely to their final values. In contrast, starting with e.g. a value of 1 for both β would lead to a ‘confused fit procedure’ that might not converge. With this approach of providing meaningful start values, fitting has proven to be very stable and robust for the 240 s batches. Here it shall be referred to the response to your next remark, where we will demonstrate that the batch size in which the g_2 data is processed has a distinct influence on the fit quality and robustness.

How does the averaging of the data (here over $\Delta t = 240$ s) effect the outcome of the fits in respect to the robustness of the factorization approach? If the window (i.e., Δt) is wider or more narrow, do the fitting results remain robust and comparable to the here presented outcome?

We thank the referee for the question, which goes hand in hand with his/her previous remark and also the question about the transit decorrelation. We tested different Δt , obtaining mostly comparable results. The choice of the optimized Δt is dictated by the necessity to be shorter than any transit effect (described above) and to be able to observe as much as possible a full decorrelation at low temperatures in order to measure accurately the compressed decay.

In the following, we compare 120 s and 240 s batching on hand of the cooling scan, please see Fig. R11. Both batching sizes provide very similar results, but especially β_s is way more scatter-loaded. This result is expected, since the shorter data range provides less information to determine the shape of the compressed component, and hence, the fitting error increases. Finally, we determined 240 s as the best option. We intentionally avoided larger batch sizes to stay away from transit decorrelation influences and keep the smearing effect on an acceptable level, as discussed above.

Figure R11: A comparison of KWW_{MULTI} fit results for 120 s and 240 s batch size, demonstrated on hand of the XPCS cooling scan.

A question on some small details: What is the detector width, or the width of the q -range assumed as being “single q ” at 2.86 \AA^{-1} ?

The q range captured by the detector is 0.3 \AA^{-1} . The area of the detector was divided in three ROI, each of one corresponding to a width q of 0.1 \AA^{-1} . The middle ROI containing $q_{\max}=2.86 \text{ \AA}^{-1}$ was selected for the analysis. In the materials and methods section, the information is accordingly added, please see the highlighted text.

Along the KWW fitting procedure, the values for the baseline b and the early plateau c are evaluated where accessible, and an average based on these values is fixed for data sets where this limits are not resolved. By how much do these values fluctuate? It might be interesting to see the evolution of these values with temperature and scanning direction.

Thank you for the comment. Please find below in Fig. R12 an example of the b and c determination process (on hand of the heating scan data). The blue data points show the average of the first three g_2 data points of each 240 s batch (which are a good measure of the early plateau). The green ones describe the average over the last three data points, describing the baseline value. We see that the first average values stay on a stable plateau before the dynamics already within the three first 0.01 s frames. Inversely, the baseline can be clearly seen at higher temperatures, but is not any more reached at lower temperatures. Hence, meaningful values for the initial plateau and the baseline are fitted (solid lines) to allow to determine b and c .

Furthermore, we observed no differences in b and c between heating and cooling.

Figure R12: Averages of the first and last three g_2 data points of the 240 s evaluation batches. The initial plateau and the baseline are determined through fits in the respective regions, allowing to define fixed values for b and c .

Reviewer 3

Thank you very much for your valuable feedback. We think your points lead to interesting discussions that we develop below.

The paper uses powerful 4th generation synchrotron radiation to investigate the dynamics of a metallic glass former. The authors should be commended on their significant advances compared to previous synchrotron XPCS measurements; achieving measurement times of 10ms compared to seconds. However, the scientific impact is not at all clear.

Until now, the general accepted idea was that the density fluctuations probed by XPCS corresponded to the microscopic α -relaxation process which was still locally active in the glassy state, although controlled by a different mechanism of particle motion with respect to the sub-diffusive dynamics observed in supercooled liquids. Our work shows that a second independent process instead sets in at the glass transition temperature, faster than the main structural relaxation, and that dominates the response in the deep glass. This interpretation modifies completely the interpretation of the glass transition and the modelling of the intermediate scattering functions. In this scenario, the α -relaxation is still partially active below T_g , modifying the shape of the density-density correlator at short time scales.

Considering that similar dynamical crossover (stretched-to-compressed evolution of the ISF on entering a gel/glassy state) are common in many complex soft materials (microgels, colloidal suspensions, clays ...), and that although two decades of studies an interpretation is still missing, we believe that our study provides new important information for both the Material Science and Soft Matter community.

The equilibrium versus nonequilibrium arguments made are confusing and incorrect. The supercooled liquid is refereed to an equilibrium state throughout the manuscript. Supercooled liquids are by definition metastable, as is the glassy state.

Indeed, the supercooled liquid (SCL) is in a metastable equilibrium state. When we refer to it as an equilibrium state, we mean a state in a relative minimum of the potential energy landscape. The term ‘metastable equilibrium’ then further specifies that this minimum is not global, but local, hence there exists a crystalline state with even lower potential energy (‘stable equilibrium’). Differently, the glass is a ‘liquid that has fallen out of equilibrium’ according to Austen Angell³⁷. The glass is then identified not as a metastable equilibrium state but as a non-equilibrium state. These considerations are well-summarized by the Figure R13, which is taken from reference³⁸.

[REDACTED]

Figure R13: A scheme that illustrates the entropy, enthalpy, volume, and the theoretical position in the potential energy landscape of (a) the metastable equilibrium SCL, (b) the unstable non-equilibrium glass, and (c) the stable equilibrium crystal. Taken from ³⁸.

In our community, this terminology is of standard use, but it may lead to misunderstandings. We tried to clarify the notation by including some text in the first line of the introduction.

Two different regimes of particle motion are assumed in modeling the supercooled liquid and glassy states. Why only two, could there be more than two ? Heterogeneous liquid like atomic motions corresponding to the alpha-relaxation process have already been observed using XPCS.

The supercooled liquid is usually addressed to (sub-)diffusive motions, and the glass is often addressed to ballistic (super-diffusive) motions. This was reported in the early XPCS studies on metallic glasses ^{24,39-45}. As we investigate the glass transition, i.e. the transition from the SCL to the glass, we focus on these two types of motion. This is the first time when these two contributions to the g_2 function are taken into account and not separately as it was in the past.

What is different here is the interpretation of the glassy state. Vitrification is described as the interlocking of nm-scale rigid domains with slower dynamics. But the suggestion of spatial heterogeneities and presence of medium range order in this metallic glass is not supported by any evidence and is tenuous at best.

Spatial heterogeneity is a broadly established concept, which we introduce in the first segment of the discussion section by citing a large fundus of literature:

“The shape exponents β and β_s in Fig. 3C and F feature values distinctly below unity that reflect the stretched exponential decay typical for the heterogeneous nature of supercooled metallic glass forming liquids^{2,18,23,24}. Furthermore, β and β_s decrease with decreasing temperature, as previously observed in the isothermal XPCS studies by N. Neuber et al.². While the origin of this well-known temperature trend in the shape exponent is still subject of vital debates²⁸, it is often correlated with liquid dynamics becoming more temporally^{25–27} and spatially^{20,21,27} heterogeneous with undercooling. While the former describes a general, non-localized tendency towards a broadening relaxation time distribution, $D(\tau')$ (see Eq. 5), the latter specifically refers to spatial fluctuations in the dynamics^{21,29}. P. Voyles et al. recently used novel electron correlation spectroscopy (ECS)³⁰ to image such spatio-temporally heterogeneous dynamics in supercooled $\text{Pt}_{57.5}\text{Cu}_{14.7}\text{Ni}_{5.3}\text{P}_{22.5}$, an alloy similar to our present system^{31,32}. Large differences in relaxation time of up to two orders of magnitude were observed between neighboring nm-sized domains, thus, on a length scale that is typical for the medium range order (MRO). This implies a sub-diffusive^{25,33,34} structural relaxation process that is governed by cooperative atomic rearrangements and caging effects.”

Here, we again want to highlight the electron correlation spectroscopy (ECS) study by P. Voyles et al.³¹, as it images spatio-temporally heterogeneous dynamics in supercooled $\text{Pt}_{57.5}\text{Cu}_{14.7}\text{Ni}_{5.3}\text{P}_{22.5}$, a system that is very similar to our present alloy. Fig. R14 provides an example of their results.

[REDACTED]

Figure R14: An example of spatio-temporal heterogeneous dynamics in supercooled $\text{Pt}_{57.5}\text{Cu}_{14.7}\text{Ni}_{5.3}\text{P}_{22.5}$, imaged via ECS. Taken from³¹.

Most metallic glasses have little to no medium range order as indicated by the presence of a first sharp diffraction peak. Is there even a pre-peak in the structure factor of this glass at low-Q that could support your model ?

Metallic glass formers are widely known for their explicit medium-range order, as written in manifold articles. Please find some here ^{46–48}. Our present Pt_{42.5}Cu₂₇Ni_{9.5}P₂₁ alloy also features a distinct pre-peak, as demonstrated in ^{49,50}.

We hope these considerations address your concerns.

References

1. Yanagishima, T., Russo, J., Dullens, R. P. A. & Tanaka, H. Towards Glasses with Permanent Stability. *Phys. Rev. Lett.* **127**, 215501 (2021).
2. Neuber, N. *et al.* Disentangling structural and kinetic components of the α -relaxation in supercooled metallic liquids. *Commun. Phys.* **5**, 1–10 (2022).
3. Liénard, F., Freyssingéas, É. & Borgnat, P. A multiscale time-Laplace method to extract relaxation times from non-stationary dynamic light scattering signals. *J. Chem. Phys.* **156**, (2022).
4. Parisi, D. *et al.* Static and dynamic properties of block copolymer based grafted nanoparticles across the non-ergodicity transition. *Phys. Fluids* **32**, (2020).
5. Jain, A. *et al.* Three-step colloidal gelation revealed by time-resolved x-ray photon correlation spectroscopy. *J. Chem. Phys.* **157**, (2022).
6. Marques, F. A. D. M. *et al.* Structural and microscopic relaxations in a colloidal glass. *Soft Matter* **11**, 466–471 (2015).
7. Física, C. De, Cfm, D. M., Upv, C. & Materials, E. H. U. Dielectric Susceptibility of Liquid Water: Microscopic Insights from Coherent and Incoherent Neutron Scattering. *Phys. Rev. Lett.* **117**, 1–5 (2016).
8. Arbe, A. *et al.* Collective dynamics and self-motions in the van der Waals liquid tetrahydrofuran from meso- to inter-molecular scales disentangled by neutron spectroscopy with polarization analysis. *J. Chem. Phys.* **158**, (2023).
9. Leheny, R. L. XPCS: Nanoscale motion and rheology. *Curr. Opin. Colloid Interface*

- Sci.* **17**, 3–12 (2012).
10. Angelini, R. *et al.* Glass-glass transition during aging of a colloidal clay. *Nat. Commun.* **5**, 1–7 (2014).
 11. Ladd-Parada, M. *et al.* Using coherent X-rays to follow dynamics in amorphous ices. *Environ. Sci. Atmos.* **2**, 1314–1323 (2022).
 12. Jankowski, M. *et al.* The complex systems and biomedical sciences group at the ESRF: Current status and new opportunities after extremely brilliant source upgrade. *Nucl. Instruments Methods Phys. Res. Sect. B Beam Interact. with Mater. Atoms* **538**, 164–172 (2023).
 13. Narayanan, T., Chèvremont, W., Zinn, T. & Meneau, F. Small-angle X-ray scattering in the era of fourth-generation light sources. *J. Appl. Crystallogr.* **56**, 939–946 (2023).
 14. Lu, X., Mochrie, S. G. J., Narayanan, S., Sandy, A. R. & Sprung, M. How a liquid becomes a glass both on cooling and on heating. *Phys. Rev. Lett.* **100**, 5–8 (2008).
 15. Ricci, A. *et al.* Intermittent dynamics of antiferromagnetic phase in inhomogeneous iron-based chalcogenide superconductor. *Phys. Rev. B* **101**, 1–6 (2020).
 16. Zhang, Q. *et al.* Dynamic Scaling of Colloidal Gel Formation at Intermediate Concentrations. *Phys. Rev. Lett.* **119**, 1–6 (2017).
 17. Schroers, J. & Johnson, W. L. Highly processable bulk metallic glass-forming alloys in the Pt-Co-Ni-Cu-P system. *Appl. Phys. Lett.* **84**, 3666–3668 (2004).
 18. Amini, N. *et al.* Intrinsic relaxation in a supercooled ZrTiNiCuBe glass forming liquid. *Phys. Rev. Mater.* **5**, 1–8 (2021).
 19. Nagel, S. R., Angell, C. A. & Ediger. Supercooled Liquids and Glasses. *Journal of Physical Chemistry* vol. 100 13200–13212 (1996).
 20. Ediger, M. D. Spatially heterogeneous dynamics in supercooled liquids. *Annu. Rev. Phys. Chem.* **51**, 99–128 (2000).
 21. Richert, R. Heterogeneous dynamics in liquids: Fluctuations in space and time. *J. Phys. Condens. Matter* **14**, (2002).
 22. Götze, W. & Sjögren, L. Relaxation processes in supercooled liquids. *Reports Prog. Phys.* **55**, 241–376 (1992).

23. Ruta, B. *et al.* Relaxation dynamics and aging in structural glasses. *AIP Conf. Proc.* **1518**, 181–188 (2013).
24. Hechler, S. *et al.* Microscopic evidence of the connection between liquid-liquid transition and dynamical crossover in an ultraviscous metallic glass former. *Phys. Rev. Mater.* **2**, 1–6 (2018).
25. Arbe, A., Colmenero, J., Monkenbusch, M. & Richter, D. Dynamics of glass-forming polymers: “Homogeneous” versus “heterogeneous” scenario. *Phys. Rev. Lett.* **81**, 590–593 (1998).
26. Faupel, F. *et al.* Diffusion in metallic glasses and supercooled melts. *Rev. Mod. Phys.* **75**, 237–280 (2003).
27. Ballesta, P., Duri, A. & Cipelletti, L. Unexpected drop of dynamical heterogeneities in colloidal suspensions approaching the jamming transition. *Nat. Phys.* **4**, 550–554 (2008).
28. Cangialosi, D., Alegría, A. & Colmenero, J. On the temperature dependence of the nonexponentiality in glass-forming liquids. *J. Chem. Phys.* **130**, (2009).
29. Richert, R. Perspective: Nonlinear approaches to structure and dynamics of soft materials. *J. Chem. Phys.* **149**, (2018).
30. He, L., Zhang, P., Besser, M. F., Kramer, M. J. & Voyles, P. M. Electron Correlation Microscopy: A New Technique for Studying Local Atom Dynamics Applied to a Supercooled Liquid. *Microsc. Microanal.* **21**, 1026–1033 (2015).
31. Zhang, P., Maldonis, J. J., Liu, Z., Schroers, J. & Voyles, P. M. Spatially heterogeneous dynamics in a metallic glass forming liquid imaged by electron correlation microscopy. *Nat. Commun.* **9**, 1–7 (2018).
32. Chatterjee, D. *et al.* Fast surface dynamics on a metallic glass nanowire. *ACS Nano* **15**, (2021).
33. Chaudhuri, P., Berthier, L. & Kob, W. Universal nature of particle displacements close to glass and jamming transitions. *Phys. Rev. Lett.* **99**, 2–5 (2007).
34. Weeks, E. R., Crocker, J. C., Levitt, A. C., Schofield, A. & Weitz, D. A. Three-dimensional direct imaging of structural relaxation near the colloidal glass transition. *Science (80-.).* **287**, 627–631 (2000).

35. Busch, S., Jensen, T. H., Chushkin, Y. & Fluerasu, A. Dynamics in shear flow studied by X-ray Photon Correlation Spectroscopy. *Eur. Phys. J. E* **26**, 55–62 (2008).
36. Gabriel, J., Blochowicz, T. & Stühn, B. Compressed exponential decays in correlation experiments: The influence of temperature gradients and convection. *J. Chem. Phys.* **142**, (2015).
37. Angell, C. A. Formation of glasses from liquids and biopolymers. *Science (80-.)*. **267**, 1924–1935 (1995).
38. Cangialosi, D. Dynamics and thermodynamics of polymer glasses. *Journal of Physics Condensed Matter* vol. 26 (2014).
39. Zhou, H. *et al.* X-ray photon correlation spectroscopy revealing the change of relaxation dynamics of a severely deformed Pd-based bulk metallic glass. *Acta Mater.* **195**, 446–453 (2020).
40. Evenson, Z. *et al.* Comparing the atomic and macroscopic aging dynamics in an amorphous and partially crystalline Zr₄₄Ti₁₁Ni₁₀Cu₁₀Be₂₅ bulk metallic glass. *J. Mater. Res.* **32**, 2014–2021 (2017).
41. Ruta, B., Giordano, V. M., Erra, L., Liu, C. & Pineda, E. Structural and dynamical properties of Mg₆₅Cu₂₅Y₁₀ metallic glasses studied by in situ high energy X-ray diffraction and time resolved X-ray photon correlation spectroscopy. *J. Alloys Compd.* **615**, 45–50 (2014).
42. Soriano, D. *et al.* Relaxation dynamics of Pd-Ni-P metallic glass: Decoupling of anelastic and viscous processes. *J. Phys. Condens. Matter* **33**, (2021).
43. Das, A., Derlet, P. M., Liu, C., Dufresne, E. M. & Maaß, R. Stress breaks universal aging behavior in a metallic glass. *Nat. Commun.* **10**, 1–9 (2019).
44. Das, A., Dufresne, E. M. & Maaß, R. Structural dynamics and rejuvenation during cryogenic cycling in a Zr-based metallic glass. *Acta Mater.* **196**, 723–732 (2020).
45. Ruta, B. *et al.* Atomic-scale relaxation dynamics and aging in a metallic glass probed by X-ray photon correlation spectroscopy. *Phys. Rev. Lett.* **109**, 1–5 (2012).
46. Ma, D., Stoica, A. D. & Wang, X. L. Power-law scaling and fractal nature of medium-range order in metallic glasses. *Nat. Mater.* **8**, 30–34 (2009).

47. Sheng, H. W., Luo, W. K., Alamgir, F. M., Bai, J. M. & Ma, E. Atomic packing and short-to-medium-range order in metallic glasses. *Nature* **439**, 419–425 (2006).
48. Bai, Y. W. *et al.* Heredity of medium-range order structure from melts to amorphous solids. *J. Appl. Phys.* **112**, (2012).
49. Gross, O. *et al.* The kinetic fragility of Pt-P- and Ni-P-based bulk glass-forming liquids and its thermodynamic and structural signature. *Acta Mater.* **132**, 118–127 (2017).
50. Gross, O. *et al.* Signatures of structural differences in Pt-P- and Pd-P-based bulk glass-forming liquids. *Commun. Phys.* **2**, 83 (2019).

Second Rebuttal letter for

Liquid-like versus stress-driven dynamics in a metallic glass former observed by temperature scanning XPCS

We thank the Editor and the Reviewers for their work and their appreciation towards the second version of our manuscript. While most comments seem to be addressed, some issues still remain, and we clarify them in the following. We hope that the detailed answer and the new version of the manuscript will now bring the referees to support the publication of our work.

All new changes in the manuscript and the supplementary materials are highlighted in green.

Reviewer 1

The authors have addressed most of my comments and I am happy to support publication of the manuscript.

I have only two minor comments:

We thank the Reviewer for the positive evaluation of our work. Looking at the two comments, we realized that we have been not clear enough in our previous reply letter. We apologize for the confusion, and we address the remaining issues below.

1. Figure R2 looks a bit confusing. For the dark blue circles in Fig. R2C, β_C should have been fixed when it is still varying as a function of temperature. That being said, I do not think it is necessary to fix both β_C and β_S for the KWWMULTI fitting. My intention was to improve the confidence of the fit by reducing fitting parameters, even at the cost of slightly increasing the error bar of τ . Therefore, could the authors try fix β and β_S in Fig. 3 (as shown by the dashed lines)? It should not impact the calculation of η in Fig. 4.

We apologize for the confusion. The legend of the previous Figure R2 was clearly misleading. We followed the referee advise and we fit the data by fixing only the stretched shape exponent. β_C in R2C is not fixed (the term "fixed" in the legend was referring to β_S). We again show these results in the attached Figure RR1, where we have simply modified the legend to be clearer (see cyan and blue circles obtained with $\beta_S=0.3$ fixed). The figure contains also data of the KWW fitting in the supercooled liquid phase obtained by fixing β to 0.6 (red diamonds). While these

KWW results are very similar to the results obtained by the original fitting with a free β parameter (filled blue data points), fixing $\beta_S=0.3$ of the KWW_{MULTI} fit leads to τ_S values that deviate from the expected behavior and do not overlap anymore with the supercooled liquid values (see the turquoise crossed circles). We associate this discrepancy with the small but still significant temperature dependence of β_S with decreasing temperature in the probed composition which reflects increasingly dynamic heterogeneity on cooling as discussed in Ref. ¹. For this reason, we prefer to leave the figure in the main article as it is.

This discussion has been included in the new version of the manuscript, the figure can be found in the SM.

Figure RR1: KWW and KWW_{MULTI} fitting results of the cooling scan by keeping all parameters free or by fixing the stretched shape exponent β_S in the glass to 0.3 (circles) and β in the supercooled liquid to 0.6 (diamonds).

2. Would the authors consider putting Fig. R6 in the Supplemental Materials? I believe this will be very helpful to the XPCS community.

Also, in Fig. R6, it is hard to tell how the inset is a zoom-in of the detector image, probably because the color scale is different and the diffraction ring is not visible in the inset. I am also a bit surprised that the ROI is almost three detector modules big (judging by the detector module gaps), given that the ROI looks very small from the 2D SAXS pattern. I encourage the authors to double-check those technical aspects. I also encourage the authors to add the Q_x/Q_y ticks and labels so that the readers can infer the reciprocal space range where the experiment was performed.

We thank the referee for the suggestion, and we are happy to include the figure in the supplementary materials. This figure represents just a schematical illustration of the reciprocal space covered by the WAXS XPCS experiment with respect to a standard X-ray Diffraction experiment. The referee is right to be confused by the two detector sizes. In our previous reply letter, we forgot to say that the two images correspond to two different experimental set-ups. The full diffraction ring has been measured at the Material Science Powder Diffraction (MSPD) beamline BL04 at Alba synchrotron, and $I(q)$ is the corresponding integrated intensity. The WAXS-XPCS experiments are performed at energies much lower than those usually employed in a XRD experiments and as a consequence only a small portion of the reciprocal space can be covered by the detector (which has also a smaller area with respect to those used in XRD). This smaller range covered by the Eiger detector used for WAXS-XPCS at the ID10 beamline of the ESRF corresponds to the circle in panel a) and b), and the corresponding measured speckle pattern is shown in panel c). The goal of this figure is to explain that XPCS measurements cover only the top part of the FSDP.

We created a new figure, which you can find here and in the supplementary materials as well.

Figure RR2: Schematic illustration of the portion of reciprocal space covered by a WAXS-XPCS experiment in metallic glasses. In a standard high energy X-ray Diffraction experiment it is possible to measure the whole diffraction pattern produced by a metallic glass and obtain the static profile $I(q)$ in (A) by integrating the area of the detector (B). Differently, with WAXS-XPCS one can cover only a small portion of the reciprocal space. The blue circle marks the peak of the first sharp diffraction peak, at which the present XPCS studies are performed. An example of the speckle pattern measured with XPCS in these conditions is reported in the smallest inset in (C).

Reviewer 2

I do thank the authors for their detailed answers, which have indeed been very helpful and allowed me to better understand the presented work. I do recognize how thorough the authors' work is, and I appreciate that. However, I got some doubts based on an important aspect raised by reviewer 1, so that I do not support its publication in the present form.

As the authors very nicely and sincerely showed, fitting by the sum of contributing processes instead of their product also yields convincingly converging fits. This is currently not at all communicated in the manuscript and sheds a different light onto the choice of the fitting function and thus the interpretation of the resulting parameters.

The authors base the discussion of a heterogeneous structure including the alpha process and a stress mode on the divergent evolution of dynamics. These are obtained by the multiplicative

fitting approach only, while there are two approaches available that both fit the data convincingly. What proves that the proposed multiplicative fitting describes the data in a scientifically more reliable way? Why exactly should the multiplicative approach be preferred over the additive approach?

The basis for this decision-making together with the outcome from the alternative fitting approach needs to be made transparent in the manuscript.

Some arguments for this decision were given in the authors' response to reviewer 1. To me, however, the data suggests typical behavior at the glass transition, namely the freeze out of dynamical contributions given the error of the stretched-component time scales and is thus not an argument for excluding the obtained results as unphysical.

Only if these concerns are addressed convincingly, the interpretations of the findings as alpha and stress mode become meaningful.

We thank the referee for the appreciation and all the constructive work put into the review process. We agree that the discrimination between the two different model functions is not straightforward.

Looking at previous XPCS experiments and numerical simulations in particular on soft glasses, one can indeed find both approaches. As discussed in the previous response letter, the main difference between the two models is based on the different temperature evolution of the stretched component of the structural relaxation time, τ_s . This is shown in Figure RR3 (same as in the previous response letter and now also included in the SM), where we compare the two fitting models for data obtained both by heating and by cooling.

With the additive model, by construction, the two processes occur within the final decay and the stretched component can be located only at nearly equal or shorter time scales than the compressed contribution. At the probed length scale, the correlation curves show only a continuous increase in intensity at shorter time scales. The two timescales are thus not enough separated to produce a visible second decay in the data and the fit converges with similar relaxation times for both the compressed and stretched component. As a consequence, τ_s exhibits a quite low activation energy in the glass state in the heating scan, while being almost temperature independent in the cooling scan, similarly to the compressed process. While we agree with the referee that the behavior at the glass transition in the heating scan could be indeed compatible with the freeze out of dynamical contributions, we believe that the results on cooling (where τ_s is almost temperature independent) cannot be described with the same picture and

contradict also macroscopic measurements where the continuous freezing of the different dynamical contributions leads to a smoother lowering of the activation energy below T_g .

Figure RR3: Comparison of fitting results of the multiplicative approach from the main text (KWW_{MULTI}) with those from the additive approach (KWW_{ADD}), which is in detail introduced in the SM of the manuscript.

As an example, we show mechanical relaxation measurements on another metallic glass ($Pd_{20}Pt_{20}Ni_{20}Cu_{20}P_{20}$) in Figure RR4, just as an illustration of the usually observed behavior on heating and cooling protocols. In this figure, the relaxations were made during 5000 s isothermal steps, progressively changing temperature at each step, first heating up to the glass transition region, where relaxation times become shorter than 10 s, and then cooling down. Details of this type of mechanical relaxation experiment can be found, for instance, in Ref. ², and they provide essentially the same information as the viscosity measurements shown in

Figure 4. The relaxation times obtained probing different physical properties are known to not coincide in exact values of time, but they do show the same temperature behavior. As seen in Figure RR4, the material relaxation is not frozen in the glass, it just progressively departs from the extrapolated liquid behavior and becomes very slow. The relaxation of the as-cast configuration, driven out from equilibrium at higher fictive temperature, is ‘relatively’ fast, while the relaxation of the slowly cooled material is much slower. This temperature behavior strongly resembles the course of τ_S obtained from KWW_{MULTI} fitting the XPCS scans.

Furthermore, it should be noted that the majority of the works reported in literature employ the additive model to describe the decoupling between the slow α -relaxation process with respect to faster localized processes in the supercooled phase (such as the β -relaxation) and are therefore representing a different time and temperature window of the relaxation spectrum with respect to the slow dynamics of the glass state reported in this work.

Figure RR4: Relaxation times obtained from isothermal step strain measurements. The relaxation measurements were 5000 s long and after each measurement the temperature was changed to the following isothermal step. The applied strains were 0.3 % and 0.6 %. Black: Heating of the as-cast sample; Red: Cooling after reaching the glass transition region; Blue: Reheating of the slowly cooled sample; Green: Same as black symbols using 0.6 % strain; Purple: Same as red symbols using 0.6 % strain.

For all these reasons, although far from claiming a conclusive result, we believe that the multiplicative model is more appropriate to describe the current data. However, we are aware that this description is strictly related to the experimentally probed length and time scales, which are already at the limit of the capabilities of the XPCS technique. Additional data at time scales at least two orders of magnitude faster than those probed here would definitely help to resolve the shape of the correlation functions and discriminate better between the two approaches. In particular, the presence or absence in the correlation functions of a well-defined second step at shorter times will clarify the possibility to use an additive model, as indeed is the case in the literature studies in supercooled liquid.

The additive fit approach is now explicitly mentioned in the main manuscript and is described and discussed in detail in the new version of the supplementary materials.

Reviewer 3

The authors have adequately addressed my concerns so I recommend publication. I still disagree with the blanket statement and the assertions made in references 46-48 that most metallic glasses explicitly have medium range order. Most do not exhibit a pre-peak and the observed periodicity simply arises from the packing of hard spheres, rather than some sort of preferred atomic ordering, see Price et al J. Phys. Condensed Matt. 1989 1 1005 and Salmon et al Nature 2005, 435, 75. I do agree however that this point is likely beyond the scope of this particular paper.

We thank the referee for supporting the publication of our article and for the very interesting articles suggested. We agree that the definition of medium range order in glasses is not trivial and depends of course on the interatomic interactions in the system and that metallic glasses are closer to hard spheres systems than to network glasses. As that discussion was part only of the previous response letter, those references and this discussion are not reported in the main article.

Third Rebuttal letter for

Liquid-like versus stress-driven dynamics in a metallic glass former observed by temperature scanning XPCS

Reviewer #1 (Remarks to the Author):

The authors have addressed all my comments. I am happy to recommend the manuscript for publication.

We are happy to address all your concerns, thank you very much for your recommendation.

Reviewer #2 (Remarks to the Author):

First, a serious thank you for being so clear and helpful with your response to that major concern I raised.

You are welcome. We are pleased to discuss with you the in-depth concerns about the validity and interpretation of the two different approaches. Changes are highlighted in turquoise.

Please let me give a detail response to the arguments you respond with, which are the following as I perceived them from your response letter, supplementary materials, and revised manuscript:

1) The additive approach yields $\tau_s \leq \tau_c$ by construction.

- As both timescales yield very comparable evolutions with temperature (essentially $\tau_s = \tau_c$ over the full temperature range), I also see the possibility that this additive approach “caps” the true evolution of τ_s , even though it is not known.

Thank you for agreeing that the mathematical formulation of the additive approach seems to ‘force’ or ‘cap’ the timescale of the stretched part. We would like to remark that more complex functional shapes could be also explored. However, without a strong theoretical background supporting the use of such more complicated functions and after many hours of inspecting XPCS data considering different fitting strategies, we think that the multiplicative approach provides the most convincing results.

2) The additive model is mostly applied to the supercooled liquid regime to capture decoupling of alpha and beta mode.

- That is not a strong argument for or against the one or the other approach from my point of view.

3) The additive approach is slightly more scattered.

- The additive approach still captures the data convincingly and with reasonable error.

4) Based on the additive KWW-approach, the activation energy differs for heating and cooling. However, it differs strongly from what is expected based on mechanical relaxation experiments on comparable heating/cooling rates. The multiplicative KWW-approach yields a behavior for the stretched response part, that reflects the behavior from temperature-dependent mechanical relaxation experiments.

- I see this point and it struck me at first, as I think it is a very strong argument for why applying the multiplicative KWW-approach is crucial. I wonder why you do not make use of this argument in the manuscript. Currently, the manuscript refers to this point by stating that the activation energies based on the additive KWW-approach are “rather unphysical”. However, also with this argument the important question remains: Why would one expect that in the XPCS data a de-convolution of structural relaxation dynamics is needed in order to be able to observe the dynamical behavior measured by mechanical relaxation experiments? In the manuscript, the stretched dynamics was put in connection with heterogeneous liquid-like dynamics, while the compressed dynamical contribution connects to stress-driven motion. Is there a good reason why one would see the heterogeneous dynamics in mechanical experiments, but not the stress-driven part? Specifically, compressed dynamics have been observed in literature in mechanical relaxation experiments, such as in stress-relaxation experiments after small-strain excitation as in PRB 110134309 (2024), which should be referenced in the manuscript.

Concerns on my side have been addressed with care, which I appreciate. I am pleased that the authors discuss the choice of the multiplicative approach over the additive approach in the revised manuscript and included an additional section on the additive KWW-approach in the Supplementary Materials.

We agree with the reviewer’s comments, and we also thank him/her for the interesting discussion we have held during these revision rounds. The two approaches, additive and multiplicative, can be used to fit and interpret the results, but the underlying physics that each one of the models represents is very different. In our opinion, the possibility to interpret the XPCS observations by using the multiplicative model, instead of the rather more common additive model, is a main outcome of the paper. We do not claim the question is solved, but we think that our analysis provides good arguments to consider a de-convolution of two processes as a possible explanation of the XPCS observations in metallic glass-formers.

We agree with the reviewer that the main point in favor of the multiplicative model is the comparison with macroscopic relaxation, especially in the glass transition region, more than the goodness of the fitting. Although compressed stress decay was reported recently in reference [PRB 110 134309 (2024)], this result corresponds to a single material at room temperature, very far from the glass transition. During a slow cooling from the liquid state, as the one applied in our scans, compressed mechanical relaxation shapes have never been reported. In fact, the relaxation tends to become more stretched as both actual and fictive temperature decrease. Furthermore, the temperature dependence of the macroscopic relaxation timescales also supports the multiplicative approach as discussed in the manuscript.

The reviewer raises a crucial question: ‘Is there a good reason why one would see the heterogeneous dynamics in mechanical experiments, but not the stress-driven part?’. We cannot answer this in a conclusive way. In our discussion, the compressed part is visualized as a consequence of the generation of out-of-equilibrium dynamics once the liquid-like motion becomes too slow to equilibrate the structure. This can be attributed to particle dynamics driven by the building-up of unrelaxed internal stresses, but there can be other possible types of motions related to structural changes in the glass state. It is difficult to establish a relation between the particle dynamics probed by XPCS and how an external, macroscopic applied stress is relaxed. For the moment we can only say that the multiplicative model gives, at least, a microscopic timescale and a relaxation shape, corresponding to the stretched part, which can be linked to the macroscopic response. We stress again that this does not mean at all that we claim the question is solved, but we think it gives some credit to the validity of the multiplicative approach.

We added a respective text in the results section that quotes your listed reference.

Apart from these aspects, I would like to mention some points that had been less pressing previously:

1) In the abstract, line 40-41, the authors write “This work not only extends the dynamical range probed by standard isothermal XPCS, but also clarifies the fate of the α -relaxation across the glass transition and provides a new perception on the anomalous, compressed temporal decay of the density-density correlation functions observed in metallic glasses and many out-of-equilibrium soft materials.” I suggest changing “clarifies the fate” to “adds a different view”.

We changed the text accordingly.

2) I appreciate that the authors now include TTCFs in Figure 1. It would be helpful to insert color bars indicating the covered range or to make an according statement in the Figure caption.

We agree that the additional inclusion of the color bars would lead to a more complete picture. Yet, we initially chose to omit them for increased graphical clarity (each of the six TTCFs would have its own color bar, leading to a messy appearance of the already large figure). Nevertheless,

we like your idea to include the additional information, so you can now find the TTCFs with their respective color bars in the SM (Fig. S9). In the caption of Fig. 1 in the main article, we included a respective statement that links towards the SM.

3) In Figure 2, the orange and purple shading and arrows are very helpful, but the “KWW_c” region is indicated very early if one looks at the corresponding contribution and the data. This “early onset” of the KWW_c distorts the perception within which temporal range the two different contributions dominate the dynamical response. Letting the KWW_c region start at 100 s would be more realistic instead of 10 s, and could be implemented easily.

You are right, 10 s might be a somewhat early point to claim a starting domination of KWW_c. We changed the figure, now the exact crossover point where KWW_c=KWW_s serves as the actual border, which is at 57 s. The text has been edited accordingly.

4) In line 187, the authors write “KWWMULTI allows a way better description”, which is quite subjective. I suggest changing this to “KWWMULTI allows for a more accurate fit of the data”.

Thanks, we changed the text accordingly.

5) The newly added part (lines 291-304) starts with “It is important to stress that the presence of multiple dynamical contributions in the ISF can be modelled also with a sum of two KWW expressions (further termed KWWADD) ...” and would like to clarify the situation further by adding “with comparable accuracy” at the end of the here cited part.

Good point, we added the text.

6) Recently, outside the group of authors, XPCS-related work on the structural dynamics of metallic glasses in view of structural and dynamic heterogeneity (sub-diffusive transport, elastic backbone, stretched/compressed dynamics) were published, which are meaningful in context to the here presented results (Nature Communications (2024) 15:6595; Materials Today (2025) 82, 92-98) that should be referenced in the manuscript.

Thanks, we included these latest references.

7) Reference 43 of the revised manuscript is not cited correctly in its current form.

Thank you for the hint, we corrected the issue.

8) In the SM, equation (S1) describes the transition coefficient as “f”, while in Figure S7 it is referred to as “x”. Also, in line 123 in the SM, it reads “0<=f<=1” which should be “0<=f<=1”.

We intermixed the terminologies here, as we termed the transition coefficient at times as 'x' or as 'f'. Thank you for the considerate hint, we completely changed to 'x' to avoid a double-use of 'f' as f is also used as a parameter in the main text. We also changed the typo in the text.

Frey et al. presented an intriguing scientific story on the atomic scale dynamics transition during the glass-formation process of a Pt-based metallic glass former. In addition, the KWW_{multi} analysis method effectively decouples dynamics of different physical origins and can significantly benefit the XPCS community. I would like to invite the authors to consider the following suggestions that may further improve the quality of the manuscript:

(A) I believe the relaxation time scale of the system should include both the fitted time constant and the stretched exponential argument, as the latter can significantly change the shape of the function. Specifically, in H. Guo et al., PRL 109, 055901 (2012), $\tau = \tau_{\text{SE}} * \Gamma(\alpha^{-1}) / \alpha$, where α is the stretch exponential argument and τ_{SE} is the time constant from the stretched exponential fit. The authors are encouraged to consider using this model for the calculation of viscosity in Fig. 4.

In the same vein, the authors pointed out that the improved fitting quality in Fig. 2 and 3 is a direct outcome of increasing the number of fit parameters, which I very much agree. Given that β does not vary significantly over the entire temperature range especially for the stretched KWW, the authors are encouraged to re-fit the data with fixed β to improve the confidence of the fitting. In addition, it is highly likely that a linear superposition of multiple single exponentials, or one stretched exponential and one compressed exponential, could yield similar fitting quality compared to the KWW_{multi} model in the manuscript. Have the authors considered this alternative approach and how it may alter the scientific story?

(B) Regarding the flow of the paper: I would like to invite the authors to consider the possibility of moving Fig. 4 from the Discussion section to the Result section, and lead the Discussion section with the physical picture in Fig. 5. This will hopefully highlight the interesting physics and make the manuscript more stream-lined and easier to absorb. This flow should also help filter out some of the text in the Discussion section which are not directly relevant to Fig. 5.

(C) Regarding the summary paragraph: have the authors considered elaborating the significance of KWW_{multi} in decoupling dynamics in complicated solid-state systems? As the authors pointed out, the compressed exponentials cannot be expanded in stretched exponentials and vice versa. Consequentially, the fitting is unique within the KWW_{multi} framework, and each component represents a dynamic mode with a distinct physical origin. More emphasis on the technical impact and the scientific perspectives, especially in other systems or related topics, will help end the paper on a stronger note.

Other technical comments:

1. The abstract mentioned the use of modern detector and 4th generation x-ray source. The CdTe detector is indeed relatively novel but the use of high-energy x-ray beam is not fully justified in the manuscript. In addition, it is not clear that the data was collected at ESRF or ESRF-EBS. The authors are recommended to provide further clarification.

2. The authors claimed that “This non-isothermal approach can only be sparsely found in literature, and we apply it for the first time on a metallic glass former, ...”. Temperature-varying in-situ measurements are quite common in XPCS, e.g., PRL 100, 055702 (2008) and PRL 109,

165701 (2012), or in soft materials such as PRL 119, 178006 (2017) and J. Chem. Phys. 151, 104902 (2019). The authors are recommended to rephrase this statement.

3. None of the g_2 in the paper has error bars. In addition, the Method section included the calculation for the two-time correlation function, however no two-time correlation function is shown in the manuscript or SI. Given the physical picture in Fig. 5, it is likely that heterodyning can be observed in two-time correlation with sufficient statistics. The authors are encouraged to provide error bars for g_2 and showcasing some binned two-time correlation function.

4. The authors claimed that “While the signal-to-noise ratio and detector technology limited the exposure time to several seconds in earlier studies, recent advances allowed exposure times in the sub-second regime”. Given that the kHz-frame rate detector is the current industrial standard, “Recent advances” most likely means 4th-generation sources like ESRF-EBS, however it is not clear whether the data was taken with ESRF-EBS. The authors are encouraged to rephrase their statement.

5. The manuscript does not contain any detector images of the scattering feature. The detector images contain important information such as the region-of-interest (ROI) definition and the scattering rate in terms of count-per-pixel. In addition, the onset of the crystal peaks during the nanodomain formation as the authors pointed out in the manuscript would certainly be of interest to the community. The authors are recommended to provide representative scattering patterns at different stages of the glass formation and specify the ROI.

6. Fig. 1 shows that not all g_2 decay from the same point, which may be a sign for faster dynamics. Is the contrast factor f fixed for all the fittings presented in Fig. 2 and 3? This should be stated in the manuscript along with the justification as it can alter the fitting results.

7. In the discussion of the origin of the stretched exponential, it was mentioned that the stretched exponential arises from the polydispersity of the time constant, “with a distribution $D(\tau')$ stretching over several decades”. However, simply combining two dynamic modes with time constants that are orders of magnitude apart will most likely result in a double-stage decay. Could the authors provide clarification on the origin of the stretched exponential and perhaps add more references?

8. Fig. 2 discusses different regions where the stretched and the compressed dynamics are dominating. Is it possible to label these regions on the figure?

9. In Fig. 1 and 3, the (A), (B), (C) labels are blocking the data points. Replotting is recommended.

10. In Fig. 5, the suppression of KWW_s is not very clear from the illustrative drawing. Maybe the orange arrows in the supercooled state can be drawn a bit longer?

11. What is the unit of ‘1/s’ in XPCS Methods?

The authors have addressed most of my comments and I am happy to support publication of the manuscript.

I have only two minor comments:

1. Figure R2 looks a bit confusing. For the dark blue circles in Fig. R2C, β_C should have been fixed when it is still varying as a function of temperature. That being said, I do not think it is necessary to fix both β_C and β_S for the KWW_{MULTI} fitting. My intention was to improve the confidence of the fit by reducing fitting parameters, even at the cost of slightly increasing the error bar of τ . Therefore, could the authors try fix β and β_S in Fig. 3 (as shown by the dashed lines)? It should not impact the calculation of η in Fig. 4.

2. Would the authors consider putting Fig. R6 in the Supplemental Materials? I believe this will be very helpful to the XPCS community.

Also, in Fig. R6, it is hard to tell how the inset is a zoom-in of the detector image, probably because the color scale is different and the diffraction ring is not visible in the inset. I am also a bit surprised that the ROI is almost three detector modules big (judging by the detector module gaps), given that the ROI looks very small from the 2D SAXS pattern. I encourage the authors to double-check those technical aspects. I also encourage the authors to add the Q_x/Q_y ticks and labels so that the readers can infer the reciprocal space range where the experiment was performed.